



# 1 Reviews and Syntheses: Variable Inundation
# 2 Across Earth's Terrestrial Ecosystems

James Stegen[1,2], Amy J Burgin[3], Michelle H. Busch[3], Joshua B. Fisher[4], Joshua Ladau[5], Jenna
Abrahamson[6], Lauren Kinsman-Costello[7], Li Li[8], Xingyuan Chen[1], Thibault Datry[9], Nate
McDowell[1], Corianne Tatariw[10], Anna Braswell[11], Jillian M. Deines[1], Julia A. Guimond[12], Peter
Regier[1], Kenton Rod[1], Edward K. P. Bam[13], Etienne Fluet-Chouinard[1], Inke Forbrich[14], Kristin L.
Jaeger[15], Teri O'Meara[16], Tim Scheibe[1], Erin Seybold[3], Jon N. Sweetman[8], Jianqiu Zheng[1],
Daniel C Allen[8], Elizabeth Herndon[16], Beth A. Middleton[17], Scott Painter[16], Kevin Roche[18],
Julianne Scamardo[19], Ross Vander Vorste[20], Kristin Boye[21], Ellen Wohl[22], Margaret Zimmer[23],
Kelly Hondula[24], Maggi Laan[1], Anna Marshall[22], and Kaizad F. Patel[1]
[1]Pacific Northwest National Laboratory, Richland, WA, USA
[2]School of the Environment, Washington State University, Pullman, WA, USA
[3]University of Kansas, Lawrence, KS, USA
[4]Chapman University, Orange, CA, USA
[5]University of California San Francisco, San Francisco, CA, USA
[6]North Carolina State University, Raleigh, NC, USA
[7]Kent State University, Kent, OH, USA
[8]Penn State University, State College, PA, USA
[9]National Institute for Agriculture, Food, and Environment (INRAE), Villeubanne, France
[10]Rowan University, Glassboro, NJ, USA
[11]University of Florida, Gainesville, FL, USA
[12]Woods Hole Oceanographic Institution, Woods Hole, MA, USA
[13]International Water Research Institute (IWRI), Mohamed VI Polytechnic University, Benguerir, Morocco
[14]University of Toledo, Woods Hole, MA, USA
[15]U.S. Geological Survey, Washington Water Science Center, Tacoma, WA, USA
[16]Oak Ridge National Laboratory, Oak Ridge, TN, USA
[17] U.S. Geological Survey, Wetland and Aquatic Research Center, Lafayette, LA, USA
[18]Boise State University, Boise, ID, USA
[19]University of Vermont, Burlington, VT, USA
[20]University of Wisconsin, La Crosse, WI, USA
[21]SLAC National Acceleratory Laboratory, Menlo Park, CA, USA
[22]Colorado State University, Fort Collins, CO, USA
[23]U.S. Geological Survey Upper Midwest Water Science Center, Madison, WI, USA
[24]Arizona State University, Tempe, AZ, USA
Correspondence: James C. Stegen, E-mail: james.stegen@pnnl.gov; Phone: (509) 371-6763



## Abstract

The structure, function, and dynamics of Earth's terrestrial ecosystems are profoundly influenced by the frequency and duration that they are inundated with water. A diverse array of natural and human engineered systems experience temporally variable inundation whereby they fluctuate between inundated and non-inundated states. Variable inundation spans from extreme flooding and droughts to predictable sub-daily cycles. Variably inundated ecosystems (VIEs) include hillslopes, non-perennial streams, wetlands, floodplains, temporary ponds, tidal systems, storm-impacted coastal zones, and human engineered systems. VIEs are diverse in terms of inundation regimes, water chemistry and flow velocity, soil and sediment properties, vegetation, and many other properties. The spatial and temporal scales of variable inundation are vast, ranging from sub-meter to whole landscapes and from sub-hourly to multi-decadal. The broad range of system types and scales makes it challenging to predict the hydrology, biogeochemistry, ecology, and physical evolution of VIEs. Despite all experiencing the loss and gain of an overlying water column, VIEs are rarely considered together in conceptual, theoretical, modeling, or measurement frameworks/approaches. Studying VIEs together has the potential to generate mechanistic understanding that is transferable across a much broader range of environmental conditions, relative to knowledge generated by studying any one VIE type. We postulate that enhanced transferability will be important for predicting VIE function under future, potentially non-analog, environmental conditions. Here we aim to catalyze cross-VIE science that studies drivers and impacts of variable inundation across Earth's VIEs. To this end, we complement expert mini-reviews of eight major VIE systems with overviews of VIE-relevant methods and challenges associated with scale. We conclude with perspectives on how cross-VIE science can derive transferable understanding via a 'continuum approach' in which the impacts of variable inundation are studied across multi-dimensional environmental space.

## Introduction

The chemical and biological processes within terrestrial ecosystems hinge on the presence, residence time, volume, and chemistry of water. A variety of factors influence water retention, infiltration and flow, such as land surface relief, topographic slope, subsurface permeability, evapotranspiration, and human-based modifications of the landscape. Water supply is most commonly 'top down' in the form of precipitation and overland flow or 'bottom up' due to rising water tables and transient saturation in the subsurface (Smith et al. 2017). Inundation, however, may also occur from lateral inputs, as is common in tidal systems, or from upslope inputs, as in floodplains. Regardless of where water comes from, inundation occurs when the rate of water supply is greater than the rate of export via infiltration, evapotranspiration, and runoff.

Here we define inundation as occurring when there is a near continuous aqueous barrier that limits gas phase transport between the atmosphere and the subsurface. This conceptualization is inclusive of diverse conditions, spanning from extreme events such as hurricane-driven flooding to shallow short-lived overland flow across hillslopes. We define variably inundated ecosystems (VIEs) as those that experience dynamic shifts between the presence and absence of inundated conditions, at any spatial and temporal scale. Variably inundated ecosystems cover at least 5-9 million km$^2$, or 4-7% of the Earth's land surface excluding Greenland and Antarctica. These estimates are according to monthly data over



multiple decades (Zhang et al. 2017, 2021, Davidson et al. 2018), and are likely significant
underestimates as many VIEs are not resolvable by commonly used remote sensing
techniques.
Variable inundation occurs across a wide range of terrestrial ecosystems, but the factors
governing its influences are typically studied independently without cross-ecosystem
comparisons. Some examples of VIEs are hillslopes with overland flow, non-perennial streams,
floodplains and parafluvial zones, variably inundated wetlands, vernal ponds/pools/playas, tidal
systems, coastal systems impacted by storm-driven flooding, and human-engineered systems
intended to shift inundation dynamics (e.g., flood-irrigated agriculture, stormwater infrastructure,
and constructed wetlands) (**Fig. 1**). While VIEs may be classified as wetlands under the
broadest definition from the Ramsar Convention (Secretariat 2016), there is significant variation
in how wetlands are defined (Finlayson and Van Der Valk 1995) and we do not attempt to rectify
or clarify variation in those definitions. Here, when using the term 'wetland' we simply align with
the perspective that wetlands are similar to marshes, swamps, and bogs.
Inundation dynamics are changing due to increased variability and magnitudes of
precipitation and evapotranspiration, accelerated sea level rise, and human modifications to the
Earth's land surface, including an increase in extreme events (Konapala et al. 2020, Li et al.
2022a). Extreme events such as coastal flooding are increasingly frequent, and while seasonal
drying of streams is now more common (Sweet et al. 2014, Zipper et al. 2021), some streams
are shifting from non-perennial to perennial (Döll and Schmied 2012, Datry et al. 2018a) while
others have fewer no-flow days than they did historically (Zipper et al. 2021). Wetland
inundation extent, duration, and seasonal timing are also projected to be altered by climate
change (Londe et al. 2022a). Thus, the dynamics of inundation are changing in different ways
across different VIEs (Zipper et al. 2021) such that we cannot rely exclusively on historical
dynamics to predict future impacts (e.g., on species diversity) of changing inundation dynamics
(Culley et al. 2016, Quinn et al. 2018, Rameshwaran et al. 2021, Li et al. 2022b).



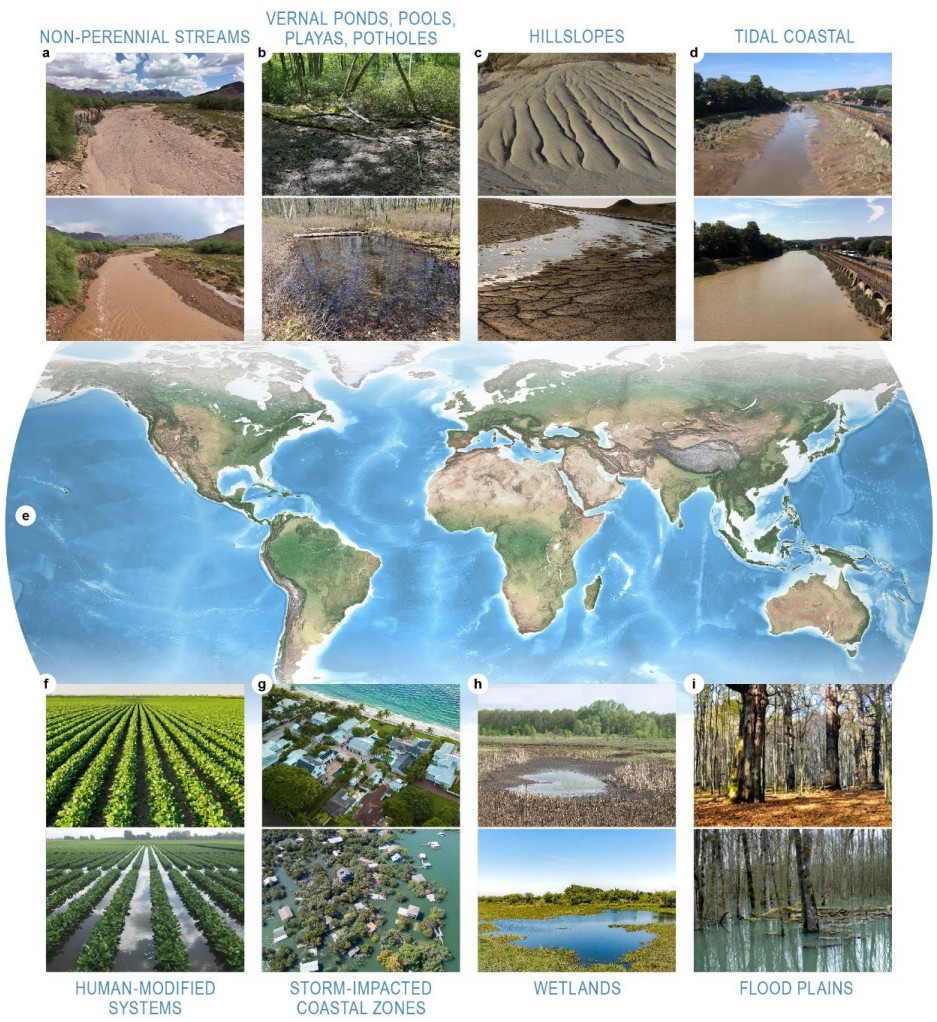

**Figure 1.** *Variably inundated ecosystems (VIEs) span numerous ecosystem types and are globally distributed across the Earth's land surface. There are few places across Earth's land surfaces that do not experience variable inundation, which is defined here as the loss/gain of an aqueous barrier between the atmosphere and porous media (e.g., soil) that inhibits gas phase transport. Due to global changes in the dynamics of variable inundation, there is a need to integrate knowledge into models that are predictive across VIEs. This will require intentionally studying VIEs together to understand how the details of any given VIE modulate the impacts of variable inundation. Credit: Nathan Johnson. There are several photos from different sources and permissions granted as follows: (a) Sullivan et al 2019; (b) Jon Sweetman, co-author; (c) Shutterstock; (d) @WeirdBristol [Twitter] 2018; (e, f, g, h) Shutterstock; (i) Mikac et al 2018.*



Mechanistic knowledge that is transferable across inundation regimes (i.e., from extreme
events to predictable cycling) and across VIEs is required to develop models that are predictive
across contemporary and future conditions. We envision the impacts of variable inundation as
dependent on the location of any given VIE in multi-dimensional environmental space. This
space can be defined with a variety of environmental variables such as inundation return
interval and duration, topographic slope, vegetation composition, precipitation, and temperature.
Many other variables could be used, but regardless, environmental change will cause VIEs to
move to different areas within multi-dimensional environmental space. Predicting future impacts
of variable inundation requires mechanistic understanding of how the location of a VIE in this
space influences those impacts. We propose that our best chance to achieve such
understanding is to generate knowledge of variable inundation impacts that is transferable
across VIEs.
Here we aim to catalyze cross-VIE science for the pursuit of transferable knowledge and
ultimately models that are predictive across and aid in conserving contemporary and future
VIEs. We briefly summarize high-level divergences in drivers of variable inundation,
commonalities in the impacts of variable inundation, and then present expert mini-reviews of
eight major VIE systems. Variable inundation occurs across vast ranges in spatial and temporal
scales, which presents challenges to cross-VIE science. As such, we overview these challenges
and offer suggested solutions along with a summary of methods that are most relevant to VIE
science. We conclude with perspectives on how cross-VIE science can derive transferable
understanding to better protect these systems and their biodiversity.



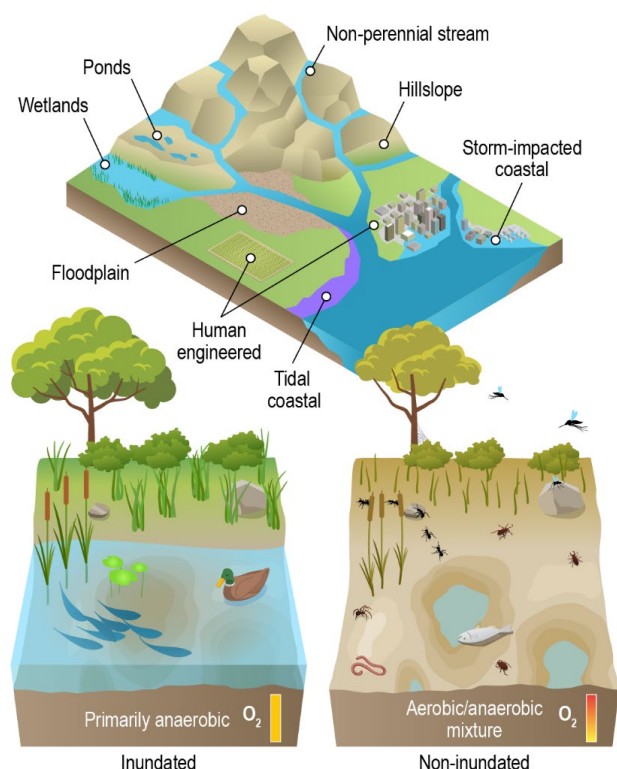

**Figure 2.** *Conceptual overview of where different types of VIEs are commonly found within watersheds and some common shifts in system states across inundated and non-inundated conditions. VIEs are found from headwaters to coastal environments (Top) and the impacts of variable inundation have some consistencies across these diverse landscapes (Bottom). Organismal ecology, physiology, and demographics are altered by variable inundation, leading to shifts in community composition. Biogeochemical processes also shift, such as greater gas-phase transport of oxygen into soil/sediment when surface water is lost, with associated shifts in redox processes. The details of these responses to variable inundation are, however, likely to vary across VIEs due to variation in system properties such as dominant vegetation types, rhizosphere development, soil/sediment texture, water salinity, flow velocity, etc. A key goal for cross-VIE science is to mechanistically link variation in these system properties to impacts of variable inundation across the multi-dimensional environmental space occupied by VIEs. Credit: Nathan Johnson.*

## Divergent Drivers, Common Responses, and VIE Mini-Reviews

The drivers of variable inundation differ markedly across VIEs and are linked to factors such as long-term drought, heavy precipitation, evapotranspiration, changing groundwater storage, soil/sediment properties, extreme climatic events, and dam operations. This leads to significant variation across VIEs in inundation regimes, which includes inundation timing, return interval,



duration, spatial extent, depth, and flow rate. For example, sediments within the active channel
of tidal rivers can experience sub-daily losses and gains of surface water, while other coastal
zones may experience extreme inundation events on a 100 year return interval. Other systems,
such as non-perennial streams and vernal ponds, also experience a broad range of inundation
regimes, ranging from sporadic and extreme inundation following rain events to more regular
seasonal cycles. Variation in the spatial scale of inundation is also large, with floodplains and
storm-impacted coastal zones experiencing inundation over tens of kilometers, whereas non-
perennial streams and ponds can experience changes across a few meters. As discussed
below within the series of VIE mini-reviews, the temporal and spatial scales of inundation also
vary substantially within each type of VIE. Variation within a given type of VIE is large enough
that we suggest it cannot be used to clearly differentiate VIEs into named categories. As
discussed in the "Toward cross-VIE transferable understanding" section, this is one motivation
for pursuing a continuum approach to cross-VIE science.
Variable inundation impacts physical [e.g., sediment transport (Peruccacci et al. 2017, Siev
et al. 2019)], chemical [e.g, water quality (Whitworth et al. 2013)], and biological/ecological [e.g.,
invertebrate communities (Plum 2005)] attributes of both natural and anthropogenically modified
ecosystems, in addition to human society (Dube et al. 2021) (**Fig. 2**). Due to intense periods of
inundation and drought, these systems are often referred to as hotspots or ecosystem control
points (Bernhardt et al. 2017), with disproportionately high reaction rates or areas of high
diversity (Davidson et al. 2012, Palta et al. 2014). In a qualitative sense, some of these impacts
are common across VIEs even if the quantitative details vary.
During inundated periods, biogeochemical processes in VIEs often shift from a dominance
of aerobic respiration during drier periods to a diverse suite of anaerobic processes, such as
methanogenesis (Datry et al. 2018b, Hondula et al. 2021b). Changes in the frequency of
inundation events change the dynamics of dry-wet, hot-cold, and aerobic-anaerobic transitions
(Valett et al. 2005). Such dynamics can challenge existing theories. For example, while rates of
soil respiration are expected to peak under aerobic conditions, periodic anaerobic conditions
can lead to unexpectedly high rates of soil carbon loss (Huang et al. 2021) and the anaerobic
process of methanogenesis can be fastest in well-oxygenated dry soils (Angle et al. 2017). More
broadly, variable inundation can alter fluxes of greenhouse gasses to the atmosphere such as
the common observation of soil rewetting leading to significant carbon loss arising from sudden
intensification of soil respiration (Schimel 2018, Shumilova et al. 2019). Variation in inundation
also has large impacts on the global $CH_4$ budget (Zhang et al. 2017, Peng et al. 2022) and
rewetting of dry sediment in intermittent streams can contribute considerably to the total $CO_2$
emissions from streams (von Schiller et al. 2019). More generally, top down and bottom up
hydrologic inundation events broadly influence biogeochemical cycles (Smith et al. 2017) and
can result in hysteretic responses to wetting and drying (Patel et al. 2022).
Across VIEs, inundation impacts the structure, composition, and function of vegetation
communities. Growth and survival can either increase or decrease with inundation depending
on local aridity and the impacts on soil hypoxia. Hypoxia kills roots, leading to reduced water
uptake, reduced photosynthesis, mortality (Pedersen et al. 2021, McDowell et al. 2022, Cubley
et al. 2023), and shifts in vegetation composition. More broadly, inundation dynamics impact
organismal ecology (Datry et al. 2023) across all VIEs, such as herbivores responding to
inundation-induced shifts in vegetation (De Sassi et al. 2012). Inundation can also alter





arthropod communities leading to reductions in diversity, abundance, and biomass with flooding
(Plum 2005). Changes at the base of food webs can have further, cascading effects (Chen and
Wise 1999).
To pursue cross-VIE science requires knowledge of the diverse array of ecosystems that
can be considered VIEs. Researchers that design and carry out cross-VIE studies may be
considered generalists in terms of the breadth of systems they study, even if their science
questions are highly specialized. To facilitate such researchers in the pursuit of cross-VIE
science, we go beyond the high-level summaries of divergences and commonalities (above)
and provide expert mini-reviews of eight primary VIE types. The following subsections present
these mini-reviews which summarize system characteristics, drivers, and impacts of variable
inundation with an emphasis on biogeochemistry and organismal ecology, and opportunities to
better understand spatiotemporal patterns and impacts of variable inundation. Each mini-review
is accompanied by a graphic that either provides a conceptual overview or imagery-based
examples, with the goal of collectively touching on key drivers, dynamics, impacts, and tangible
system examples. The collection is not meant to be a comprehensive classification of all
possible VIE types, but does cover a broad range of VIEs. The sequence of mini-reviews
roughly follows the flow of water moving from hillslopes to coastal environments (**Fig. 2**) and
includes variably inundated components of: (*i*) hillslopes, (*ii*) non-perennial streams, (*iii*) riverine
floodplains and parafluvial zones, (*iv*) wetlands, (*v*) temporary ponds, (*vi*) storm-impacted
coastal zones, and (*vii*) tidal systems. The final mini-review (*viii*) is focused on ecosystems that
have been engineered to modify inundation regimes, which occur throughout the continuum
from hillslopes to coasts.
We separate VIEs into categories as a heuristic simplification that allows for an appreciation
of variation and commonalities in drivers, impacts, and opportunities. We anticipate that the
disciplinary foci of individual researchers will align most closely with a subset of the summarized
VIE types. One goal of this manuscript is to facilitate researchers thinking about how their
science applies across VIEs. We emphasize that in many (and maybe all) cases there is not a
clear distinction among the types of VIEs we discuss below (e.g., non-perennial streams can be
flooded due to storm surge, resulting in floodplains or parafluvial zones). Ultimately, we
encourage a continuum perspective that does not rely on discrete system names or hard
boundaries, and instead views VIEs across multi-dimensional environmental space based on
inundation regimes and physical settings.
**Hillslopes with Surface Runoff**
Hillslopes provide water to lower-lying areas, often concentrating the water in gullies and
depressions (**Fig. 3**). Hillslopes produce relatively transient VIE features and may often be seen
as extensions of other VIEs, such as hillslope seeps co-located with a wetland or the
unchannelized swales that contribute to a non-perennial network. In cold regions, snow, ice and
permafrost can create an impermeable layer resulting in near-surface soil being inundated for
days to weeks during spring thaw (Coles et al. 2017, Patel et al. 2020). In dry regions, intense
precipitation that exceeds the local infiltration capacity can result in water ponding on the
surface of hillslopes or overland flow generation down hillslopes, which can be exacerbated by
initial hydrophobicity of dry soil (Kirkby et al. 2002). Exceeding the infiltration capacity is more
likely on hillslopes with low-permeability, such as clay-rich soil or when near-surface soils are



frozen. This can be exacerbated by restrictive soil horizons located at shallow depths across
hillslopes that generate seasonal perched water tables and lead to inundation (McDaniel et al.
2008). Overland flow can be spatially heterogeneous due to variations in soil characteristics as
well as flow accumulation, leading to infiltration or exfiltration along the hillslope (Betson and
Marius 1969).
In forested hillslopes, soil infiltration often exceeds rainfall intensity (McDonnell 2009, Burt
and Swank 2010) and lateral flow towards topographic depressions can lead to saturation and
ponding (Anderson and Burt 1978) (**Fig. 3a**). Microtopography within hillslopes (**Fig. 3b**) can
also lead to temporary ponding, e.g., from rain in tropical environments and from spring
snowmelt in colder environments (Clark et al. 2014). Toe slopes can generate wedges of
saturation that grow upslope (Weyman 1973, Choularton and Perry 1986), although subsurface
saturation and ponding can also occur on upper slopes where the soil is thinner [e.g., (Tromp-
van Meerveld and McDonnell 2006)]. Finally, spatial variation in topographic characteristics
(e.g., aspect, slope, curvature) can result in differences in soil moisture, incoming energy, and
vegetation, affecting evapotranspiration and inundation patterns (McVicar et al. 2007).

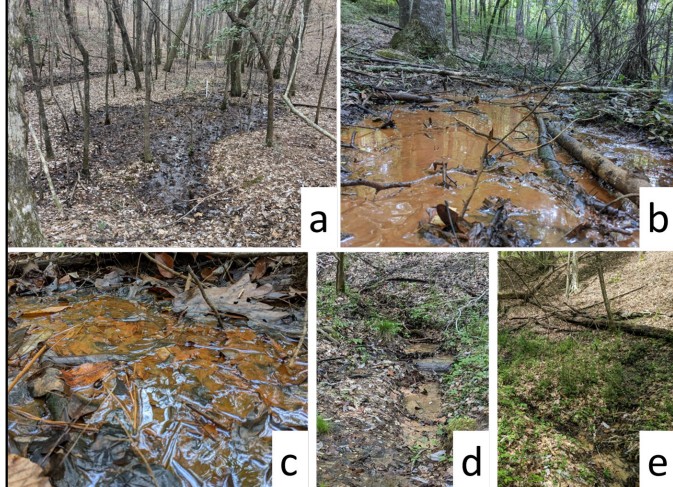

**Figure 3**. *Examples of variable inundation along hillslopes. a) looking downslope at an*
*inundated slope; b) ponding due to microtopography; c) sheet wash across the surface of a*
*hillslope; d) rill formation with turbid water from erosion; e) vegetation community change on*
*slope due to differences in soil moisture. All photos taken by Corianne Tatariw at Tanglewood*
*Forest, Alabama.*
Surface runoff and inundation on hillslopes can result in the export of soil nutrients,
salinization of soil from groundwater seeps, erosion, and landslides. There is a balance between
the effects of variable inundation on hillslope vegetation and erosion. In water-limited systems,
inundation can increase plant productivity and diversity, as well as increased rooting strength of
soils (Zhao et al. 2022) (**Fig. 3e**). However, increased inundation can also lead to increased
chemical weathering and lower shear strength in hillslope soils during storms, leading to higher
erosion and landslide potential. Along with erosion, landslides and soil compaction are inherent





to many hillslopes, which also can create areas ripe for inundation (Bogaard and Greco 2016).
At shoulder and midslope positions, increased overland flow due to saturation- or infiltration-
excess increases sediment detachment, which is then deposited in foot and toe slopes (Huang
et al. 2002). The transport of particles also leads to the transport of nutrients that are sorbed to
the particles, such as phosphorus. Erosion can be concentrated in rills and gullies or can spread
out across a slope as 'sheet wash' that impacts large areas of hillslopes (**Fig. 3c,d**). Impacts of
erosion are dependent on interactions between the persistence of inundation and soil properties
(Thomas et al. 2020).
The aqueous chemistry of water that is transported over hillslope surfaces reflects the
chemistries of contributing water sources such as precipitation, shallow soil water, and
exfiltrating groundwater. Shallow soils in hillslopes have abundant organic materials and
nutrients (Herndon et al. 2015), whereas organic matter decreases with depth, solutes derived
from the parent rock material increase with depth (Brantley et al. 2017). These stratifications
collectively regulate source water chemistry in hillslopes. Dry to wet transitions shift flow paths
from groundwater to soil water dominance in streams, therefore shaping stream chemistry (Zhi
and Li 2020, Stewart et al. 2022). Dry to wet transitions also shift water content and pore space
oxygen concentrations (Jarecke et al. 2016, Smyth et al. 2019), often triggering the release of a
cascade of solutes produced under anaerobic conditions (Schlesinger and Bernhardt 2020).
These entangled, complex interactions among hydrological and biogeochemical processes
often challenge the differentiation of individual processes and mechanistic understanding on
how variable inundation regulates flow paths, reactions, stream chemistry, and solute and gas
export fluxes (Li et al. 2021).
Investigations of variably inundated hillslopes present significant and challenging research
opportunities due to their inherently dynamic nature. One key challenge is quantifying the
occurrence and spatial extent of hillslope VIEs across the globe. Remote sensing could be used
to identify and quantify these areas, spatially and temporally, based on sky-visible vegetation
(e.g., plant morphologies, leaf nutrient contents) and topographic signatures (e.g., erosional
patterns) caused by variable inundation. To fully understand the ecological and biogeochemical
impacts of variable inundation on hillslopes, research needs to focus on shallow subsurface
physical properties, hydrology, and their linkage to biogeochemical processes. This can be
pursued via environmental geophysics to map and characterize the influence of subsurface
restrictive layers (Fan et al. 2019 p. 201). Understanding the subsurface soil architecture is key
to predicting variable inundation from bottom-up and top-down water sources, along with the
follow-on impacts to ecology and biogeochemistry.
**Non-Perennial Streams**
Non-perennial streams, defined as rivers and streams that cease to flow at some point in either
space or time (Busch et al. 2020), are ubiquitous and comprise 50-60% of the global river length
(Messager et al. 2021). These systems occur across all continents and biomes (Messager et al.
2021). Streamflow in non-perennial streams ranges from nearly perennial (year-round) flow, to
seasonal flow, responding to drivers like snowmelt, to daily or sub-daily flow events responding
to rainfall/flood events or evapotranspiration (Price et al. 2021). At the reach scale, non-
perennial streams shift between three main states - flowing, ponded/pooled, or no-surface water
present (**Fig. 4**). As reaches become hydrologically connected (or disconnected), the spatial





footprint/extent of the connected stream network can grow or shrink over sub-daily to seasonal
to interannual timescales (Xiao et al. 2019). Spatial and temporal shifts among the three
hydrologic states strongly influence the network's capacity to process, transport, and export
material to downstream systems (Allen et al. 2020).
The high variability in the spatial and temporal scales of streamflow intermittency is
indicative of the complex set of interacting drivers that induce stream drying. At the global and
regional scales, the degree of aridity is a primary control on the abundance of non-perennial
streams (Hammond et al. 2021, Zipper et al. 2021). At smaller scales, catchment properties
exert strong control over both the capacity of water delivery to the channel and the subsequent
balance between the channel and near subsurface capacity to transport water (Hammond et al.
2021, Zipper et al. 2021, Price et al. 2021). Non-perennial flow can occur anywhere in the steam
network, from headwaters to higher order rivers. While some networks display longitudinal
transitions from non-perennial to perennial flow (or vice versa), other networks exhibit more
complex patterns in surface water flow and connectivity, which may be driven by topography,
geology, vegetation, or groundwater abstraction/use (Costigan et al. 2015, 2016).
The variable inundation dynamics in non-perennial streams have cascading implications for
biogeochemical cycling, water quality, ecosystem function, and community ecology. Under non-
flowing conditions, riverbeds are characterized by dry conditions or discontinuous and stagnant
water pools, often with high temperatures, low dissolved oxygen levels, and long residence
times, functioning more like soils (Arce et al. 2019), as described also in the hillslope section.
Pooled, non-flowing conditions can lead to steep redox gradients in the shallow subsurface that
drive nutrient processing (Datry and Larned 2008, Gómez-Gener et al. 2021, DelVecchia et al.
2022). During dry/non-flowing states, terrestrial organic matter accumulates in the channel and
is subjected to varying degrees of breakdown (Datry et al. 2018c, Del Campo et al. 2021).
Rewetting of accumulated substrates can stimulate microbial activity, nutrient attenuation
(Saltarelli et al. 2022), and generate pulses of greenhouse gasses such as $CO_2$ and $N_2O$ (Datry
et al. 2018a, Song et al. 2018). During re-wetting and resumption of flow, non-perennial streams
can contain large amounts of terrestrial and aquatic organisms that can be flushed downstream
(Corti and Datry 2012, Rosado et al. 2015), with high sediment, dissolved organic carbon, and
solutes (Laronne and Reid 1993, Hladyz et al. 2011, Herndon et al. 2018, Wen et al. 2020,
Fortesa et al. 2021, Blaurock et al. 2021).
Biological responses to rewetting depend on the distribution of habitats and biota at the
watershed scale and the duration of the preceding dry phase. In highly dynamic river systems,
such as braided rivers, drying and wetting cycles can be spatially patchy and short-lived but
frequent, and thus ecological recovery following wetting can be very rapid due to the very active
hyporheic zones characterizing these systems (Arscott et al. 2002, Vorste et al. 2016). In other
systems recovery can be slow, depending on the proximity of refuges, such as springs, isolated
pools, and perennial reaches (Sarremejane et al. 2021, Fournier et al. 2023). Systems with
frequent and severe drying events are more likely to be colonized by aerial or other overland
dispersers than by aquatic dispersers (Bonada et al. 2007, Bogan et al. 2017a, Sarremejane et
al. 2021). Life-history events of some species coincide with predictable rewetting events, such
as post-snowmelt fish spawning (Hooley-Underwood et al. 2019) and amphibian and insect life
histories (Bogan et al. 2017a). Rewetting also partly determines the germination success and
establishment of riparian vegetation (Merritt and Wohl 2002).



Compared to their perennial counterparts, non-perennial streams have received less
research and monitoring attention and tend to be undervalued relative to ecological/functional
performance of perennial streams (Palmer and Hondula 2014). As such, many of the pressing
research needs in non-perennial streams are limited by data availability (Van Meerveld et al.
2020, Zimmer et al. 2022). Non-perennial streams are systematically under-represented in
global gaging networks (Messager et al. 2021, Krabbenhoft et al. 2022), leading to major gaps
in our understanding of the timing, magnitude, and duration of flow in diverse non-perennial
streams. In addition, our ability to predict the onset or cessation of flowing periods is limited by a
lack of gaging. Infrequent grab sampling for water chemistry tends to undersample non-
perennial streams specifically, leading to an even greater paucity of biogeochemical data from
these systems, particularly during rapid re-wetting events. Spatially explicit data on streamflow
intermittency and subsurface conditions at fine spatial scales (10s of meters) remain limited to a
few intensively studied catchments [e.g., (Zimmer and McGlynn 2017)]. While some global scale
datasets on streamflow intermittency have been developed (Messager et al. 2021), the
resolution of these products necessarily omit smaller, headwater reaches, hindering our ability
to quantify hydrologic and biogeochemical processes in non-perennial streams broadly
(Benstead and Leigh 2012).
Major challenges and opportunities include accurate mapping of non-perennial streams and
accurate predictions of flow timing at annual, seasonal, and shorter time scales across scales.
With limited time series data, predictions of flow in terms of duration, frequency, and spatial
extent can be challenging. How the timing and frequency of flow will change under climate
change remains an open question. It is expected that an increased frequency and duration of
droughts will shift streams toward more non-perennial flow states (Döll and Schmied 2012). In
contrast, flow permanence may increase in select areas where streams are fed by melting
glaciers or snowpack, or where anthropogenic intervention occurs (Datry et al. 2023). The
changing frequency of extreme flow events and rapid no-flow/high-flow oscillations also have
the potential to further alter streamflow, biogeochemical processes, and organismal ecology in
non-perennial streams, necessitating further integrated hydro-biogeochemical studies in these
dynamic systems.



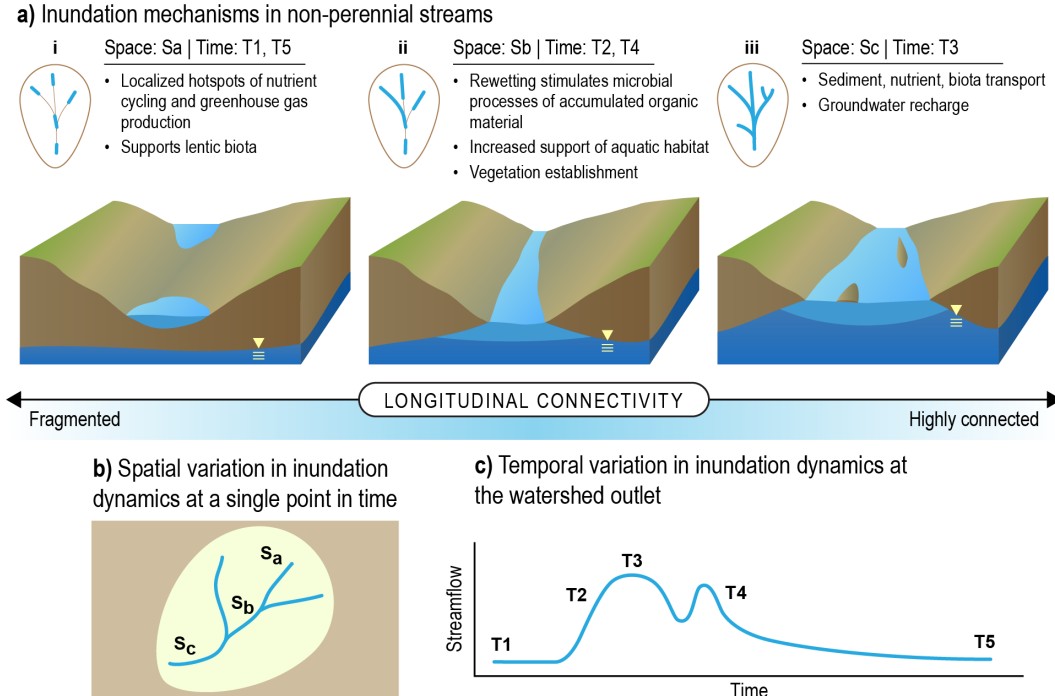


**Figure 4**. *Conceptual model of variable inundation in non-perennial streams. a) Water connections between groundwater, near surface, and surface regions at locations within a given network result in varying degrees of longitudinal connectivity with associated biogeochemical processes. b) At a single snapshot in time, water connections result in spatial variation in surface water inundation. c) Under time varying flow states, extent of surface inundation will also vary at a given location. Inundation mechanisms depicted in a) represent a losing system that is transitioning to a flowing state. We acknowledge that in some systems, a low flow fragmented state also occurs in gaining streams with locally connected groundwater. Spatial variation is signified by Sa - Sc and temporal variation is signified by T1 - T5. Credit: Nathan Johnson.*

## Floodplains and Parafluvial Zones

Rivers, both perennial and non-perennial, create two types of VIEs, floodplains and parafluvial zones (**Fig. 5**). Floodplains are alluvial landforms generated by river erosion and deposition and hydrologically connected to the contemporary active channel but outside the active river channel (Nanson and Croke 1992). Parafluvial zones are areas in the active channel without surface water at low flow, i.e., at higher-elevation areas within an active channel that contains perennial flow (Goldman et al. 2017). Nearly all rivers have parafluvial zones and adjacent floodplains, although these may be longitudinally discontinuous (e.g., absent where the river flows through a narrow bedrock gorge or descends into the subsurface). Consequently, the global distribution of these environments is extensive, as few terrestrial surfaces do not include a river network.



Spatial scales of inundation in floodplains and parafluvial zones are variable between rivers
and through time along a river. Fundamentally, spatial scales are governed by the interaction
between the magnitude of flow and available space as defined by topography. Floodplains of
the world's largest rivers such as the Amazon, Congo, or Mississippi can extend laterally for
kilometers on both sides of the active channel. In contrast, the floodplain of a headwater
channel may be only 1-2 m wide on each side of the channel.
Temporal scales of inundation (e.g., frequency, periodicity, intensity) vary substantially
across climates, topographic regions, and river network position. A snowmelt-dominated or
monsoon-fed river will have a regular annual flood that lasts for weeks, whereas a small stream
dominated by convective rainfall or tropical depressions may have irregular floods that only last
for hours. Although precipitation-driven over bank flow from the main and tributary channels is
the primary driver of inundation on floodplains and parafluvial zones, inundation also results
from direct precipitation, rising water tables, and overland flow from adjacent uplands (Mertes
2011). Thus, inundation of floodplains may be directly related to their proximity to variably
inundated hillslopes and streams.
The nature of floodplain/parafluvial inundation affects the dynamics of surface and
subsurface water, solutes, particulate organic matter, sediment, and biota (Junk et al. 1989).
Dynamics include volume and duration of storage; rate of movement; direction of movement
between surface, hyporheic, and groundwater; and biogeochemical alterations that in turn
impact river water quality, greenhouse gas emissions, plant function, and organismal ecology.
The duration, frequency, and areal extent of floodplain/parafluvial inundation control ecosystem
function, and the types and abundances of organismal communities, including both aquatic and
terrestrial species (Ward et al. 1999). Species distribution, movement, and biological
interactions, such as predator-prey, are intricately tied to these inundation patterns (Robinson et
al. 2002, Stanford et al. 2005). Fish species, for example, can migrate from dry season refugia
into floodplains during inundation, influencing food web structure and ecosystem productivity
(Crook et al. 2020).
Among the primary challenges to answering questions regarding the variation in
floodplain/parafluvial inundation are limited monitoring data and a lack of numerical models that
integrate knowledge across disciplines and processes. Measurements and models of hydrology
commonly treat floodplains as flat, impermeable surfaces, which ignores surface-subsurface
water exchanges that influence hydrology and ecosystem function (Wohl 2021). Models also
often ignore the micro-heterogeneities that influence spatially and temporally variable patterns
of inundation, biogeochemical cycling, and ecology in both floodplains and parafluvial zones.
The degree of physical detail represented in models often involves tradeoffs in spatiotemporal
extent; a one-dimensional model might ignore microtopography that influences important
inundation details, whereas a more representative two-dimensional or three-dimensional model
becomes computationally intensive for larger spatial extents. This problem gives rise to the
challenges and opportunities for (i) designing measurement campaigns across disciplines that
can create integrative data for diverse floodplains and parafluvial zones to adequately represent
the physical complexity of variable inundation processes at broad scales, and (ii) developing
floodplain/parafluvial functional groups [e.g., (Fryirs and Brierley 2022)] that can facilitate
understanding of scaling and transferability of data.



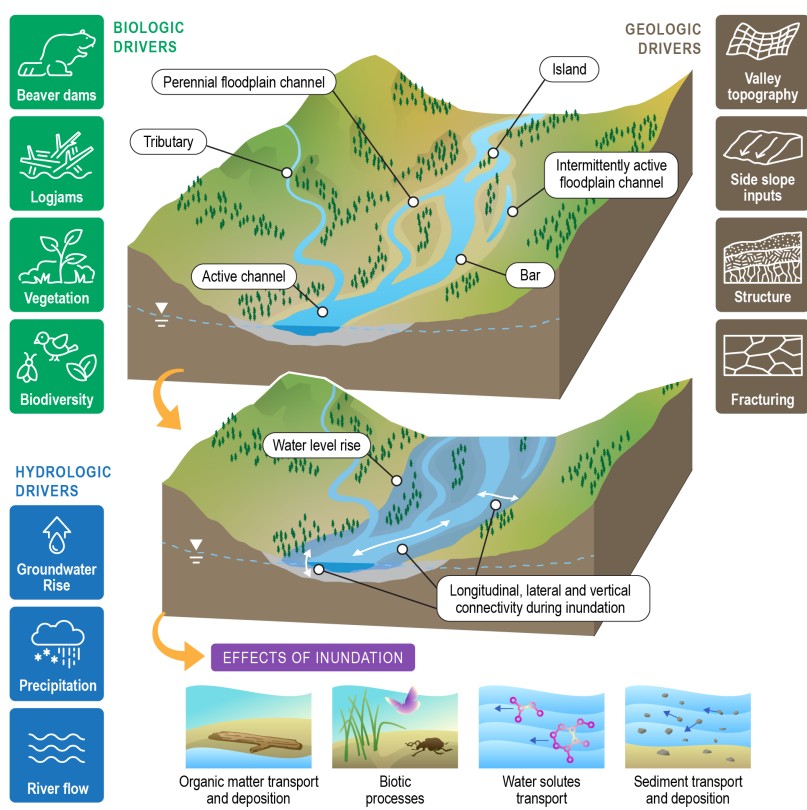

**Figure 5**. *Conceptual model of variable inundation in floodplain and parafluvial systems.*
*Across floodplains and parafluvial zones a suite of biological, hydrologic, and geologic factors*
*drive inundation regimes in terms of spatiotemporal duration, timing, depth, flow rate, etc. These*
*systems include diverse subsystems as summarized in the top panel. Rising water levels, due*
*to one or more drivers, can inundate these subsystems as shown in the middle panel, resulting*
*in a variety of biogeochemical, ecological, and physical effects (bottom sub-panels). Credit:*
*Nathan Johnson.*
**Variably Inundated Wetlands**
While not all wetlands are variably inundated, variable inundation is a common feature of many
wetland ecosystems [e.g.("Convention on wetlands of international importance espeically as
waterfowl habitat" 1994), US 33 CFR § 328.3]. Here we focus primarily on wetlands that are
similar to swamps, marshes, and bogs (**Fig. 6**). Wetlands cover about 10% of the global land
area, and nearly half of global wetland area (46%) is temporarily inundated (Davidson et al.
2018). Generally, wetland inundation regimes are shaped by the wetland's connectivity to
surface and subsurface hydrologic sources and landscape position (Åhlén et al. 2022). The
landscape position of wetlands is a first order indicator of the water source and chemistry,
ranging from headwater depressional locally-fed wetlands, to flow-through and fringing wetlands





to groundwater-fed low-lying wetlands (Fan and Miguez-Macho 2011, Tiner 2013). Wetland
typologies applied in several national inventories generally rely on a combination of three
criteria: soil type, hydrophytic vegetation and hydrology (Cowardin and Golet 1995).
Alternatively, hydrogeomorphic classification systems propose to exclusively draw on physical
drivers, such as geomorphology, hydrology and substrate to allow for cross-site comparisons of
biota and serve functional assessments (Brinson 1993, Semeniuk and Semeniuk 1995, 2011,
Davis et al. 2013).
While changes to inundation extent and depth can occur at time scales ranging from days to
decades, the most conspicuous inundation patterns occur on event (e.g., flooding due to rain
events), seasonal (e.g. snow melt or wet/dry seasons), and interannual time scales. Primary
drivers of inundation in unmanaged wetlands come from subsurface groundwater discharge and
surface flows including rainfall or snowmelt runoff that occur when antecedent soil moisture
conditions are high, preventing quick infiltration of water (Rasmussen et al. 2016). Many
wetlands are actively managed, such as to provide bird habitat, so that inundation can vary
based on management decisions [see below and (Fredrickson and Taylor 1982)].
The spatial scales of variable inundation are shaped both by wetland size and
geomorphology. Wetlands can be shallow over large spatial scales, and thus the size of variably
inundated wetland area can range from microtopographic (i.e., hummock/hollow, ~$m^2$ scales) to
larger ecosystem scales. Large wetland areas, especially in the tropics, experience strong
seasonal inundation cycles which depend on changes in water balance and local topography
(Zhang et al. 2021). While the largest variably inundated wetlands are connected to floodplains,
like the 130,000 $km^2$ Pantanal (Ivory et al. 2019), non-floodplain wetlands surrounded by upland
(also known as geographically isolated wetlands) as large as ~6 ha may also experience whole-
system drying and rewetting (Lane and D'Amico 2016).
Embedded within wetland ecosystems, microtopographic structures can create within-
system mosaics of inundation regimes. Microtopography in peaty wetlands is particularly
pronounced, ranging from several tens of meters [e.g., ridges and sloughs (Larsen et al. 2011)]
to meters [e.g. hummock-hollows (Shi et al. 2015)], These spatial patterns result from dynamic
feedbacks between ecological processes (e.g. peat accumulation) and hydrology that reinforce
these patterns (Belyea and Baird 2006, Eppinga et al. 2008, Larsen et al. 2011).
Wetlands are widely acknowledged to be biogeochemical hot spots and ecosystem control
points (McClain et al. 2003, Bernhardt et al. 2017) because of the confluence in space and time
of allochthonous substrates into reactive environments (e.g., nitrate produced under oxic
conditions entering anaerobic environments where denitrification can occur). In addition,
variable inundation is associated with nutrient influx into wetlands that replenishes nutrient pools
and can drive productivity and organic matter decomposition (Venterink et al. 2002). The depth
and duration of flooding shapes the wetland vegetation community by controlling germination
success, modifying oxygen availability and changing concentrations of toxins and nutrients, by
desiccating aquatic plants or inundating terrestrial plants, and by changing the light availability
(Casanova and Brock 2000). Wetland vegetation is structurally adapted to low oxygen
environments, for example, some vegetation has developed air channels in leaves, stems, and
roots to transport oxygen belowground (Tiner 2017). Alternatively, wetland vegetation can also
respond to shifts in oxygen levels physiologically on shorter time scales (Colmer 2003).
Variable inundation provides an environmental filter for biota adapted to live either under dry
or inundated conditions, resulting in distinct communities including wetland obligate and



facultative species (Gleason and Rooney 2018). The temporal duration of inundation (i.e.,
hydroperiod) indirectly controls the bird community composition through absence and presence
of wetland vegetation and availability of aquatic macroinvertebrate prey (Daniel and Rooney
2021). Amphibian communities are particularly impacted by hydroperiod: It needs to be long
enough for eggs to hatch and tadpoles to reach metamorphosis, but should not allow the
establishment of many predator species (Resetarits 1996).
Predicting how complex inundation patterns in wetlands will change under changing climate
is a major research challenge. Predictions span the range from a decrease in inundation in
some regions (Londe et al. 2022b) to an increase in others (Watts et al. 2014), with uncertain
consequences for wetland persistence overall. To improve regional or global predictions,
accurate maps of wetland extent on different scales that can be incorporated into mechanistic
models will be necessary (Melton et al. 2013). This is particularly challenging for non-permanent
wetlands, which are hard to reliably map and are generally understudied (Gallant 2015, Calhoun
et al. 2017), but which are, by definition, VIEs. As climate change alters wetland inundation
regimes, the net impacts to carbon storage and greenhouse gas fluxes are of particular concern
(Moomaw et al. 2018), because together they will determine the net climatic impact of changes
in wetland area and dynamics (Neubauer and Megonigal 2015).

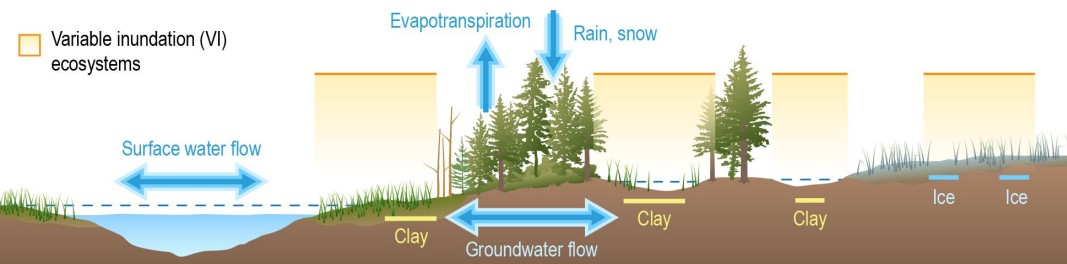

**Figure 6**. *Conceptual model of variable inundation in wetland systems. Different wetland*
*types are influenced and shaped by variable inundation. Absence and presence of surface*
*water is driven by (e.g., seasonally) changing water supply and the hydrologic function of the*
*wetland in the landscape. Sediment characteristics (e.g., clay or ice) and topographic positions*
*of wetlands in the landscape influence water loss to infiltration or gain from groundwater. Credit:*
*Nathan Johnson.*

### Freshwater Ponds

Freshwater ponds are among the most abundant and common freshwater ecosystems
worldwide, with estimates between 500 million and 3.2 billion ponds globally (Davidson et al.
2018, Hill et al. 2021). Ponds are generally small (less than 5 ha) and shallow (less than 5 m),
and consequently, are highly sensitive to changes in water levels that can result in highly
variable inundation regimes (Gendreau et al. 2021, Richardson et al. 2022a). Pond ecosystems
are extremely diverse, and include arctic thermokarst ponds, prairie potholes, vernal pools,
playas, rock pools and agricultural dugouts. The numbers of ponds globally are likely
underestimated, as their size and ephemeral/temporary nature has meant they are often





excluded from physical inventories and they are below the resolution of many remote sensing
techniques (Hayashi et al. 2016, Calhoun et al. 2017, Hill et al. 2021).
As in many other VIEs, inundation of freshwater ponds can be highly variable, and the
timing, duration and frequency of inundation can vary considerably (Williams 2006). Many
temporary or ephemeral ponds can become intermittently or seasonally flooded (**Fig. 7**). For
some ponds, particularly vernal pools, seasonal inundation is relatively predictable, as these
systems become inundated following snowmelt or spring runoff, and are subsequently drawn
down with increasing summer evapotranspiration (Zedler 2003, Brooks 2004). Variation in the
hydroperiod can alter the composition of biotic communities (Brooks 2004, Gleason and Rooney
2018), as well as impact biogeochemical and hydrological processes (Bam et al. 2020, Hondula
et al. 2021b). In more temperate regions, the timing of inundation is often driven by heavy
rainfall, and periods of inundation can be highly variable, with inundation durations lasting from
days to months, and sometimes occurring intermittently as ephemeral systems dry and rewet
multiple times in a year (Ripley and Simovich 2009, Kneitel 2014, Florencio et al. 2020). For
nearly permanent ponds, the pattern of wet and dry periods are more predictable, but the
initiation and length of the hydroperiod can vary spatially as water levels fluctuate, inundating
and exposing shallower areas (Brendonck et al. 2017). Freshwater ponds often demonstrate
both high inter- and intra-annual variability, and diurnal, annual and multidecadal periods of
inundation can occur due to changes in evapotranspiration, drought, drainage, flooding, and / or
hydrologic function of the pond on the landscape (Brooks 2004, Gendreau et al. 2021).
Modifications to ponds by humans (e.g. irrigation ponds, urban stormwater ponds; see section
on human-engineered systems) or other organisms, such as beavers, can also impact
hydroperiod and inundation regimes (Renwick et al. 2006, Brazier et al. 2021).
Like many of the other ecosystems that experience variable inundation, freshwater ponds
are also considered biodiversity and biogeochemical hotspots, providing many critical
ecosystem services (Capps et al. 2014, Marton et al. 2015). Despite their relatively small size,
ponds can have considerable variability in both community composition and in biogeochemical
processes, in part due to differences in inundation regimes, where pond margins are more likely
to be more frequently desiccated for longer periods than central regions (Reverey et al. 2018).
Models that explicitly incorporate remotely sensed variable inundation predict that ephemeral
systems with shorter hydroperiods retain nitrogen at greater rates than larger systems with less
variable inundation and longer hydroperiods, particularly in semi-arid regions like the Prairie
Potholes of the North American northern Great Plains and playas in the south-central United
States (Cheng et al. 2023). In addition, research suggests reproduction is largely impacted by
inundation. Salamanders, for example, tend to lay more eggs during years with greater rainfall
while hatching success decreases with desiccation (Della Rocca et al. 2005). Variable
inundation across ponds can result in ecosystem heterogeneity at the landscape scale,
increasing local abiotic and biotic variation (Jeffries 2008), but the number and distribution of
inundated ponds can also impact regional biodiversity through processes like dispersal
(Brendonck et al. 2017).
Climate change will likely alter the inundation regimes in freshwater ponds in terms of timing,
frequency, duration, and extent. Decreases in precipitation and increases in extreme drought
can result in shortened hydroperiods, and increasing temperatures can alter water temperatures
and evaporation rates (Matthews 2010). The persistence of freshwater ponds may, therefore, be



reduced with climate change (Londe et al. 2022b). Understanding how future changes in
inundation regimes impact freshwater ponds will be critical. Similar to wetland ecosystems,
improved remote sensing methods, including incorporating multispectral imagery and radar
along with finer spatial resolution mapping approaches may improve the mapping, counting and
inclusion of small ponds in freshwater inventories (Bie et al. 2020, Rosentreter et al. 2021,
Hofmeister et al. 2022). As inundation regimes may become more variable, increasing
conservation and protection efforts for ephemeral and temporary ponds may become more
essential to maintain these critical VIEs.

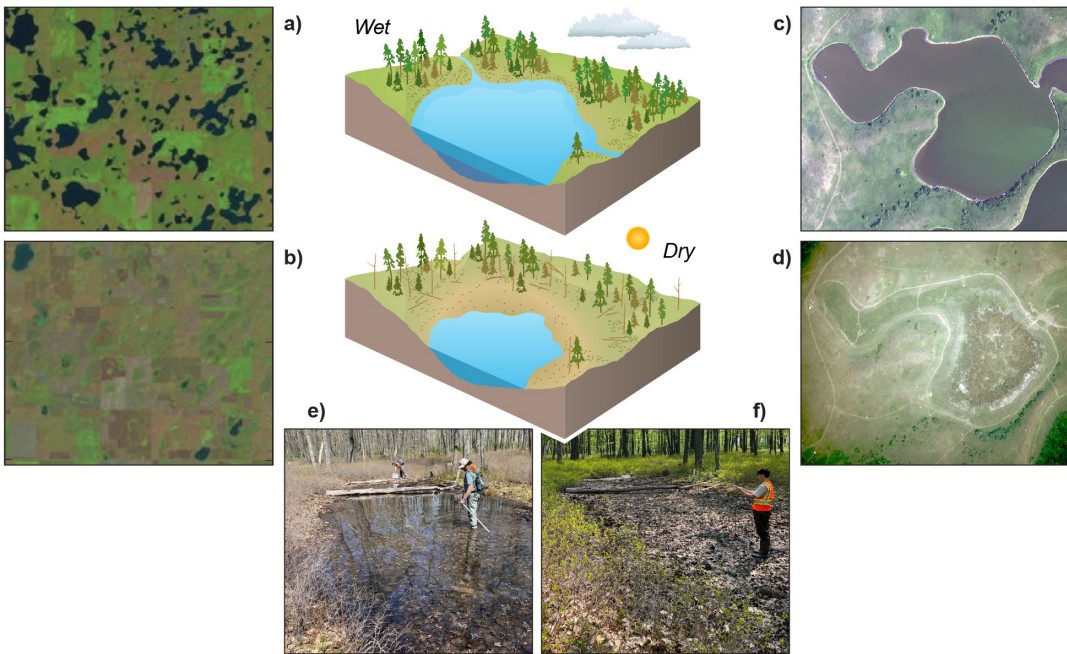

**Figure 7.** *Examples of variable inundation across scales in pond systems. Satellite*
*imagery of the Prairie Pothole Region, North Dakota, USA illustrating decadal variable*
*inundation at  a landscape scale a) September 2, 1992; b) May 23, 2013 [modified from*
*(Scientific Investigations Report 2015)] and at the pond scale;  Aerial Imagery of Pond P1,*
*Cottonwood Lake Study Area, North Dakota c) September, 2002 d) September, 1992 (Images*
*from (U.S. Geological Survey 2017). Seasonal changes in a vernal pond in Moshannon State*
*Forest, Pennsylvania, USA) inundated (May 11, 2023) non-inundated (May 23, 2023) (J.N.*
*Sweetman). Conceptual drawings by Nathan Johnson.*
**Storm-Impacted Coastal Zones**
The coastal zone includes ecosystems and communities (cities/towns) that are adjacent and
hydrologically connected to a large water body (e.g., ocean, Great Lakes). These systems
influence, are impacted by, and are dependent on coastal zone hydrologic processes, such as
flooding, that occur at the interface between terrestrial and aquatic domains. Unlike tidal





environments, inundation that affects the coastal zone is driven by temporary, often stochastic
events including storms, seiches, and king tides. Depending on the topography of the area,
infrastructure of the community, and size of the event, the size of coastal inundation varies from
event to event (both geographic impact and aerial extent of inundation; **Fig. 8**). The frequency of
these events ranges from multiple times a season to decadal (**Fig. 8**). Tropical storms and
cyclones develop in tropical regions during seasonal periods of warm water each year. Due to
their high energy and movement, they influence more temperate regions as well (Colbert and
Soden 2012). In temperate or cold regions, storms develop in the winter time due to large
temperature differences between land and ocean (Liberato et al. 2013). Natural systems will
display some form of resilience and recovery to storm impacts (Lugo 2008, Wang et al. 2016),
but human settlements and infrastructure are vulnerable to both intense winds and flooding
(Lane et al. 2013, Hinkel et al. 2014, Braswell et al. 2022). Land use development also alters
the natural resilience of coastal environments through the proliferation of gray infrastructure
such as jetties and seawalls (Gittman et al. 2015). Systems in low-lying regions are particularly
vulnerable to inundation as opposed to rocky shores with steep slopes. While regional or global
data sets based on elevation data exist, the extent at any given time of storm surges, king tides,
and other high water episodes depend locally/regionally on where the event hits, flooding
infrastructure, and topography of the area.

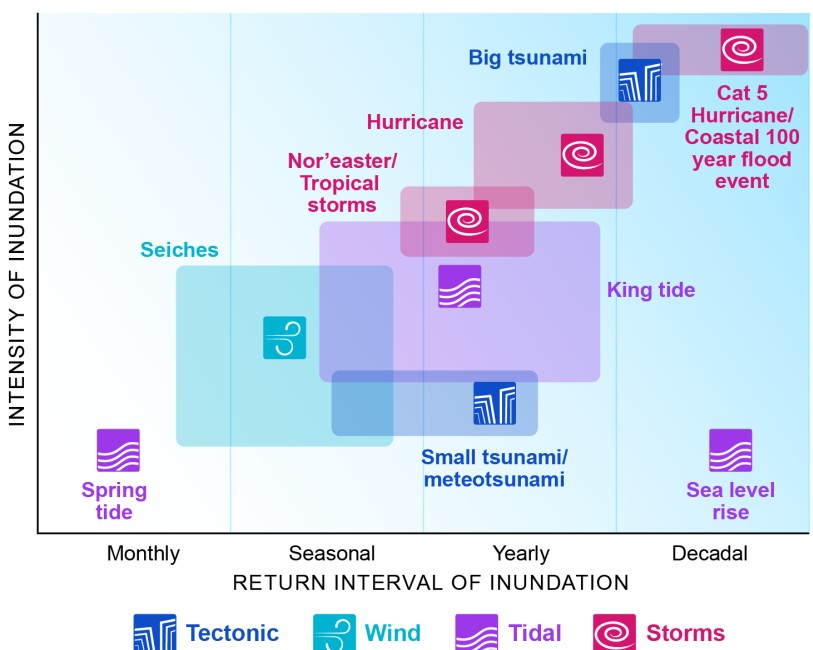

**Figure 8**. *Coastal VIEs experience inundation events with different frequencies and*
*intensities. Some events occur rarely, but are very high intensity events (category 5 hurricanes;*
*large tsunamis), increasing the area of inundation and affecting areas that seldom experience*
*flooding. The impacted systems are often less adapted to inundation, increasing the extent of*
*destruction or reorganization of the system. Other events occur more regularly and/or are lower*






*in intensity (spring tide, seiches), leading to less extensive inundation and impacting coastal*
*systems that are more adapted to inundation. Credit: Nathan Johnson.*

Inundation in the coastal zone impacts sediment transport, solute and nutrient mobilization,
vegetation distribution, biological diversity, and biogeochemical processes. Erosion and
sediment deposition alter ecosystem geomorphology (e.g., dune shape, marsh accretion)
(Houser and Hamilton 2009, Dissanayake et al. 2015) and ecosystem nutrient pools [e.g.,
(O'Mara et al. 2019, Castañeda-Moya et al. 2020)]. In coastal zones adjacent to marine and
estuarine waters, saltwater intrusion changes surface (Schaffer-Smith et al. 2020) and
groundwater (Cantelon et al. 2022) quality and mobilizes nutrients through porewater ionic
exchange processes (Herbert et al. 2018). Coastal zone inundation as a natural process alters
dune systems, which generates a mosaic of habitats that increase biodiversity (Smith et al.
2021) and alter distributions of vegetation and animals. For example, the frequency of overwash
events affects plant composition and diversity on sand dunes (Stallins and Parker 2003) and
regular inundation is thought to provide necessary habitats for some insects and birds (Smith et
al. 2021). Increased salinity and associated geochemical changes alter microbial community
diversity and population heterogeneity (Nelson et al. 2015), shifting to more specialized
communities as an adaptation to anaerobic conditions, redox fluctuation, and salt stress.
Previous studies found high variability in relationships between salinity and ecosystem carbon
dioxide fluxes (Morrissey and Franklin 2015, van Dijk et al. 2015, Dang et al. 2019, Hopple et al.
2022).

Human communities within the coastal zone are impacted by inundation events as well.
Inundation of coastal agricultural lands from storm surge and sea level rise reduces agricultural
productivity (Lei et al. 2016). In particular, risk is high to coastal zone communities in developing
nations, where inundation events can lead to food insecurity, loss of livelihood, and increased
transmission of waterborne diseases. As climate change alters the magnitude and frequency of
inundation in the coastal zone, it will be necessary to integrate both natural and human
adaptations, such as enabling salt marsh transgression (marsh migration upland) to mitigate
storm surge impacts on crop yield (Guimond and Michael 2021).

While we understand many of the linkages between the ecology, biogeochemistry,
hydrology, and geomorphology that regulate ecosystem structure and function in coastal
systems (Fagherazzi et al. 2012, Hinshaw et al. 2017, Braswell and Heffernan 2019, Cantelon
et al. 2022), we know little of how to predict the future effects of the interacting stressors
associated with climate change (O'Meara et al. 2017, Ward et al. 2020, Arrigo et al. 2020). Our
ability to predict is reliant on our understanding of shifting inundation regimes in the context of
elevated $CO_2$, nutrient pollution, and coastal development which can generate antagonistic,
synergistic, or additive effects. These knowledge gaps stem from the dynamic and
unpredictable nature of events that drive coastal inundation. Observational data to inform
mechanistic models is limited and governed by where and when events happen (not necessarily
within monitored sites), funding periods, and accessible coastlines. This difficulty is exacerbated
by the fact that 40% of the world's population lives within 100 km of the coast (Maul and Duedall
2019), which heightens social impacts of variable inundation while also adding logistical
difficulty to coastal monitoring. When events do overlap with instrumented sites, the extreme
nature of inundation events threaten instrumentation arrays, risking washout or flooding of





monitoring infrastructure. Lastly, high-latitude coastlines are also susceptible to coastal
inundation, yet few models incorporate physical, biogeochemical, and ecological implications of
inundation on permafrost bound coastlines and environments (Ekici et al. 2019, Bevacqua et al.
2020). Opportunities of critical knowledge advancement exist in 1) monitoring events through *in-*
*situ* or remotely sensed monitoring data, 2) model development that integrates more robust
process-based understanding, and 3) expansion into urban and permafrost-bound coastlines.
**Tidally Driven Coastal Zones**
Tidally-influenced coastal zones exist at the intersection of terrestrial and marine environments
and encompass diverse intertidal ecosystems including tidal wetlands, flats, and beaches (**Fig.**
**9**). Globally, tidal wetlands exist on 6 of 7 continents, and are spread across tropical, temperate,
and polar latitudes (Murray et al. 2022a). Tidal flats are predominantly found along low sloping
coastlines with approximately 70% of global tidal flat area existing in Asia, North America, and
South America (Murray et al. 2022b), while beaches encompass 31% of ice-free shorelines
(Luijendijk et al. 2018).
Tidally-driven coastal zones are inundated semi-diurnally (i.e., twice a day) or diurnally (i.e.,
once a day). Unlike VIE systems summarized above, where inundation events may be difficult
to predict, inundation in tidally-driven coastal zones varies primarily based on predictable
drivers. For example, high tide and low tide water levels dictate the spatial extent and duration
of inundation. In addition, intra-annual tidal dynamics are largely controlled by lunar cycles
which drive approximately monthly highest (spring) and lowest (neap) tides, as well as annual
high (king) and low tides. Inter-annual tidal dynamics are linked to sea level rise, which is
shifting the zone of variable inundation inland (Ensign and Noe 2018, Tagestad et al. 2021). We
note that while the timing of king tides is predictable (perigean spring tide), their impacts can be
difficult to predict, as mentioned in the storm-impacted coastal zones section. In addition,
topography (e.g., slope) and other natural physical factors, including wind speed and direction,
waves, and even localized high and low pressure events mediate the lateral extent of surface
water inundation in tidal ecosystems. Human modifications further alter both vertical and
longitudinal extent of tidal flooding via control structures which may exclude tides (gates, weirs,
etc.) and channels that transport tidal waters well inland of the natural intertidal zone.
The extent of tidal influence, which spans microtidal (< 2 meter tidal range) to macrotidal (>
10 meter tidal range in some locations), controls water quality, terrestrial-aquatic interactions
and resulting biogeochemical and ecological responses [e.g., (Tweedley 2016)]. Estuaries,
where tides mix saltwater and freshwater, are dynamic biogeochemical mixing zones
characterized by sharp chemical gradients that regulate biological activity [e.g., (Crump et al.
2017)]. Shifts in tidal zones associated with sea-level rise are predicted to alter the extent of key
intertidal habitats, with potential disruptions to coastal food webs (Rullens et al. 2022). Changes
in duration and extent of inundation associated with tides control soil saturation and salinity,
which influence redox dynamics, and hydrologically driven exchange of carbon, nutrients, and
pollutants (Pezeshki and DeLaune 2012, Bogard et al. 2020, Regier et al. 2021). Biological
activity, including crab burrows that alter hydrologic flow paths (Crotty et al. 2020), also
influence tidal exchanges across the coastal terrestrial-aquatic interface (Crotty et al. 2020).
Increased saltwater exposure due to shifting tidal ranges can alter the stability of coastal soils
[e.g, (Chambers et al. 2019)], which represent a globally important carbon sink (Mcleod et al.



2011). In addition, tidal regimes structure vegetation gradients, where salt-sensitive
communities including low-lying forests and freshwater marsh species are replaced by salt-
tolerant communities including mangroves and saltmarsh species (Kirwan and Gedan 2019,
Lovelock and Reef 2020). This shift in tidal range leads to the creation of ghost forests (Kirwan
and Gedan 2019), which can impact coastal biogeochemical cycles [e.g., (Cawley et al. 2014).
Similarly, sea level rise may lead to mangrove or marsh retreat as inundation patterns change
(Xie et al. 2020).
Due to the frequency of inundation, tidally inundated ecosystems are hydrologically,
biogeochemically, and geomorphologically dynamic, creating challenges for scientists and land
managers seeking accurate estimations of land surface area, elevation, and carbon storage.
These challenges are exacerbated by sea level rise, which exerts heterogeneous and non-linear
influences on tidal ranges (Du et al. 2018). Methodological approaches to assess tidal
ecosystem area and elevation that are based on satellite imagery will be critical for present and
future management and decision making. Similarly, complex feedbacks across three-
dimensional physical space exist among hydrology, biogeochemistry, ecology, and
geomorphology (Xin et al. 2022); these dynamics may need to be considered in future
ecosystem projections. Thus, a deeper understanding of feedbacks and their variability in space
and time in response to tidal activity is needed (Ward et al. 2020). Lastly, with sea-level rise,
tidal constituents may change, with nonlinear impacts on tidal range and inundation extent
(Pickering et al. 2017). Tidally inundated VIEs represent the interface between marine and
terrestrial ecosystems, and to predict their future will require understanding bi-directional
connections among physical, chemical, and biological system components.

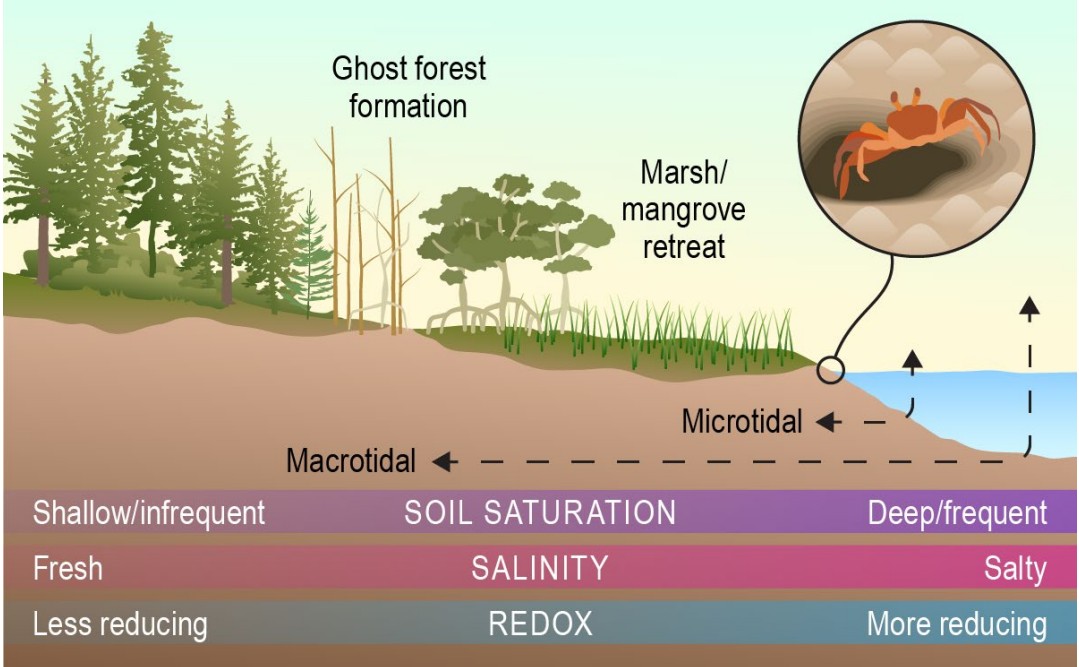




**Figure 9**. *Conceptual model of variable inundation in tidal systems. Tidally driven coastal*
*zones span sediments exposed at low tide to marshes and coastal forests inundated at high*
*tide. This lateral gradient of tidal exposure is characterized by gradients in vegetation and soil*
*characteristics, and modified by the physical, chemical, and biological factors discussed in the*
*tidal systems section. Credit: Nathan Johnson.*
**Human-Engineered Systems**
Human-engineered systems are environments where inundation magnitude, frequency, timing,
and duration are either actively managed or have been dramatically altered by structural
modifications to the landscape (**Fig. 10**). Human-engineered VIEs rival natural systems in area
and extent (Clifford and Heffernan 2018), yet the significance of engineered VIEs in influencing
landscape processes is relatively unexplored compared to natural systems (Koschorreck et al.
2020) and they are historically excluded from water and nutrient budgets (Abbott et al. 2019).
The primary drivers of human-engineered VIE formation explored here are land use change and
restoration, though hydrologic modifications impact inundation regimes of the natural VIEs
explored earlier in the manuscript. Examples of land-use driven human-engineered VIEs
include, but are not limited to, croplands irrigated by flooding (e.g., rice paddies, cranberry
bogs); irrigation and drainage canals, stormwater control structures (e.g., roadside ditches,
retention ponds), as well as unintentional VIE formation following landscape modification such
as "accidental" urban wetlands (Palta et al. 2017) or ponding in agricultural fields (Saadat et al.
2020). Whereas the purpose of land-use driven engineered VIEs is to redistribute water for
human purposes, the goal of VIEs engineered for restoration is to either replace or enhance
ecosystems lost or damaged as a result of human activity. VIE restoration efforts vary in scope
and form, spanning local (e.g., residential living shorelines, individual stream reaches,
agricultural ditch wetlands) to ecosystem [e.g., adding sediment to degrading marshes
(VanZomeren et al. 2018)], to regional (e.g., dam removal) scales.
While the full extent of human-engineered VIEs is difficult to quantify, key examples highlight
their significance in the landscape. Agriculture covers nearly 40% of the earth's land surface
(Siebert et al. 2010), and nearly a quarter of that is variably inundated by flood irrigation (Wu et
al. 2023). In urban systems, the extent of stormwater control networks rival those of natural
systems. For example, the total linear length of residential canals in North America nearly
equals that of the Mississippi River (Waltham and Connolly 2011). While restoration efforts are
not as widely distributed as land-use change, restoration still contributes to extensive VIE
creation. For example, restoration accounts for 14% of areal gain of tidal wetlands globally
(Murray et al. 2022b). Inundation regimes in human-engineered VIEs can be driven by natural
hydrologic processes, such as connectivity with the water table or tidal inputs. This is
particularly important in VIEs built for restoration, as establishing natural inundation regimes
enhances landscape connectivity and mediates ecosystem functions (Reis et al. 2017, Jones et
al. 2018). However, unlike the previously discussed natural systems, the drivers and duration of
inundation in human-engineered VIEs may be decoupled from natural hydrologic processes.
Controlling drainage, such as for stormwater management, land reclamation, or effluent
releases, is a key motivation for VIE construction and system design, resulting in inundation
periods largely driven by precipitation that persist at event to seasonal scales depending on
local hydrology and climate. Inundation duration may also occur on longer timescales, such as



seasonal flooding in paddy systems (De Vries et al. 2010). Finally, direct human interventions,
such as floodgates, weirs, and dams, may affect water residence time at timescales that are
asynchronous from natural drivers, such as seasonality or tides.
Human-engineered VIEs fundamentally alter the landscape, changing the spatial and
temporal patterns of ecosystem processes. Agricultural inundation, such as flood irrigation or
ponding, alters redox conditions, greenhouse gas emissions, groundwater recharge,
evapotranspiration fluxes, plant growth, and pollutant export to natural water bodies (Hale et al.
2015, Pan et al. 2017, Pool et al. 2021, Buszka and Reeves 2021). For example, a recent study
showed that variably inundated depressions in agricultural fields can account for ~30% of
nitrous oxide emissions across cultivated areas despite comprising ~1% of the land surface
(Elberling et al. 2023). The creation of drainage canals increases waterborne carbon fluxes from
VIEs by producing a newly decomposed stock of labile soil carbon to be leached as well as by
increasing the hydrological runoff rate through the soil and receiving canals and ditches (Stanley
et al. 2012). Human-engineered VIEs can also provide ecosystem services that supplement or
replace those of natural VIEs in the landscape (Clifford and Heffernan 2018). For example, they
can enhance habitat (Connolly 2003, Herzon and Helenius 2008), nitrogen removal (Bettez and
Groffman 2012, Reisinger et al. 2016), and recreation (Beckingham et al. 2019). Further, the
services these systems provide can be improved through targeted management [e.g.,
vegetation composition; (Castaldelli et al. 2015)] or restoration practices [i.e., two-stage ditches;
(Speir et al. 2020)].
Including human-engineered systems in our conceptualization of VIEs emphasizes the
growing significance of these systems as human landscape modifications continue to alter and
eliminate natural VIEs. Recent efforts have synthesized the role and impacts of human-
engineered VIEs at large scales (Peacock et al. 2021, Li et al. 2022b) but, as with many natural
systems, the majority of studies on human-engineered VIEs are based in North America and
Europe (González et al. 2015, Zhang et al. 2018, Bertolini and da Mosto 2021). Thus, our
knowledge may not reflect the social, political, and economic challenges of developing areas
where the highest rates of VIE modification are occurring (Wantzen et al. 2019). The knowledge
gaps surrounding human-engineered VIEs will become increasingly important to address as
global change continues to alter the spatial and temporal patterns of inundation. Given that
human-engineered VIEs can enhance or disrupt hydrologic connectivity, they potentially
magnify the effects of human driven changes such as sea level rise and impacts of
contamination from anthropogenic "chemical cocktails" (Kaushal et al. 2022). We lack a
baseline standard for how human-engineered VIEs function in the landscape, even as global
change continues to shift existing baselines [e.g., (Palmer et al. 2014)]. Addressing these
knowledge gaps will require the incorporation of human-engineered VIEs into large-scale
synthesis and modeling efforts, particularly those that address hydrologic and biogeochemical
fluxes. Conclusive definitions and inventories of human-engineered VIEs is essential for
estimating their ecological and biogeochemical roles at the global scale. Finally, human-
engineered VIEs need to be conceptualized within an ecological, rather than managerial,
context for integration and comparison with natural systems. Human-engineered VIEs rival the
range of natural VIEs in structure, inundation regime, and global distribution. Understanding
their role in the Earth system is, therefore, critical for understanding both the impacts of and
potential solutions to global change.




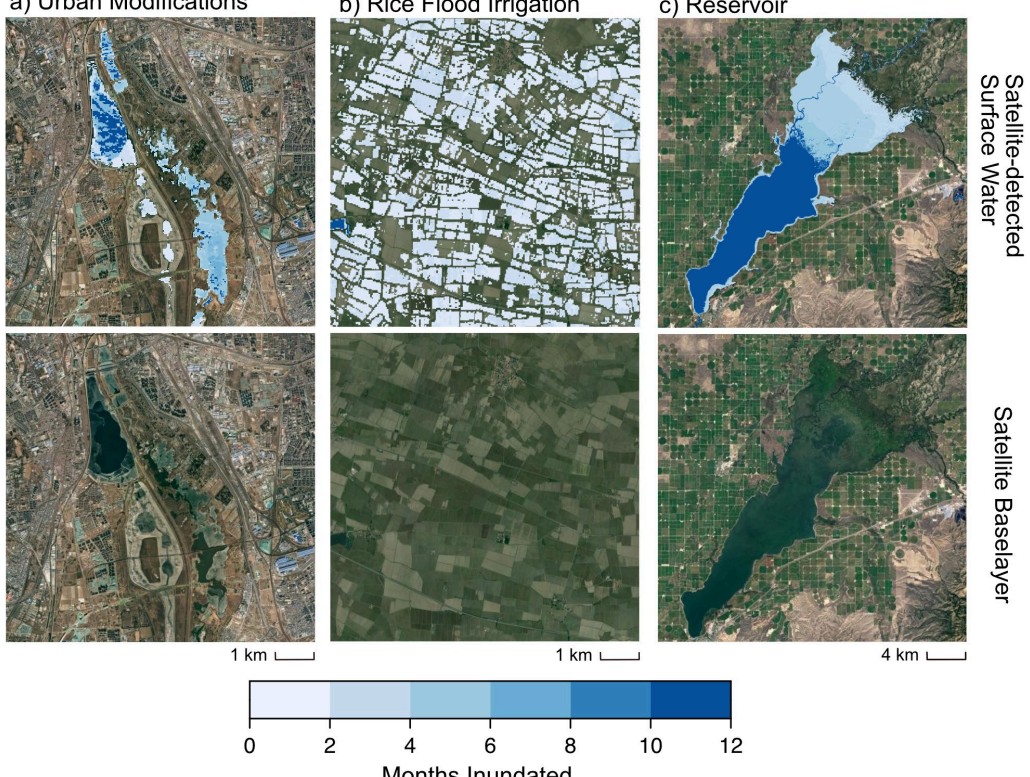


**Figure 10.** *Examples of human-engineered Variably Inundated Ecosystems. a) Yongding
River in Beijing, China; b) Paddy rice fields in northern Italy; c) American Falls Reservoir on the
Snake River in Idaho, United States. These three examples emphasize significant variation in
the degree of variable inundation across human-engineered VIEs, with some regions being
perennially inundated. Top row: Satellite-derived map data on months inundated is derived from
the "seasonality" product in the Global Surface Water Mapping Layers v1.4* (Pekel et al. 2016)*.
Credit: Jillian Deines.*

## Inundation Processes are Relevant at the Scale of the Beholder

VIEs span broad spatiotemporal scales of variable inundation, from microenvironments like
mosses and pore spaces that are periodically covered by droplets of water, to vast endorheic
lakes and rivers. Inundation volumes and surface areas of VIEs vary by at least sixteen orders
of magnitude, from under $10^{-3}$ L to over $10^{13}$ L (https://www.k26.com/lake-eyre-papers-lake-
eyre-basics), and $10^{-6}$ m$^2$ to over $10^{10}$ m$^2$ (https://www.guinnessworldrecords.com/world-
records/92443-largest-ephemeral-lake), respectively. The duration of inundation varies by up to
eight orders of magnitude, spanning a few seconds, in the case of droplets, to decades, in the
case of endorheic lakes, and centuries in the case of sea level rise. Non-inundated periods



likewise span seconds to centuries and longer. This variability in spatial and temporal extent has
profound consequences for the ecology and biogeochemistry of VIEs. This section highlights
the importance of considering scale and explores hypotheses regarding how scale drives
variability in drivers, processes, and impacts across VIEs and how we study them.

Spatial and temporal scales of VIEs can be categorized along two axes – extent and
granularity. Extent comprises the total size of the spatial domain or time duration of a defined
system, while granularity pertains to the spatial or temporal intervals of system transitions
(Ladau and Eloe-Fadrosh 2019). For example, the dynamics of water droplets across North
America would represent a large extent with fine granularity, relative to the inundation dynamics
of a several square meter desert playa (smaller extent but coarser grain). The impacts of
variable inundation are dependent 'on the scale of the beholder' relative to the extent and grain
of variable inundation, where a 'beholder' may be a molecule, organism, population, community,
land manager, or otherwise. The expressed metabolism of an individual microbe will be
influenced by inundation down to the spatial scale of water films and on hourly or shorter time
scales. An individual microbe may not, however, be influenced by whether variable inundation
occurs only within a square meter or across many square kilometers because it does not
perceive these larger scales. In contrast, macroinvertebrate behavior is influenced by variable
inundation down to scales of meters and days, and is likely further influenced by larger and
longer scales of stream network connectivity (Bogan et al. 2017b, Sarremejane et al. 2017).

VIEs can be viewed as habitat patches of different sizes that vary in how long they persist in
a given state and that have dynamic connectivity among patches. Terrestrial and aquatic biota
respond on ecological and evolutionary time scales to the expansion and contraction cycles of
inundation (Bornette et al. 1998, Ward et al. 2002). Biotic diversity is influenced by productivity,
connectivity, disturbance severity and disturbance frequency, all of which operate at hierarchical
scales (Ward et al. 1999). Biogeographical and ecological theories posit that patch size (e.g.,
species area scaling) and disturbance regimes (e.g., intermediate disturbance hypothesis) are
strong determinants of community composition (Adler et al. 2005, Svensson et al. 2012),
suggesting that VIE community composition may vary predictably with these factors. The
duration, predictability, and frequency of inundation likely have consistent community-level
consequences that vary predictably with VIE extent and grain. Different extents and grains of
inundation have the potential to change habitat connectivity in addition to directly selecting for
different groups of organisms. Isolated marshes may, for example, become merged during a
flood, thereby enhancing dispersal of aquatic organisms. The scale of variable inundation has
numerous influences over ecological processes and dynamics that need to be understood.

From a biogeochemical perspective, variable inundation generates spatial and temporal
variation in rates and patterns of biogeochemical processes. This variability is important for
scaling biogeochemical rates because of process nonlinearity and Jensen's inequality (Ruel and
Ayres 1999). That is, a rate based on average conditions differs systematically from the average
rate across variable conditions. This is important because the scales of processes (e.g.,
microbial activity occurring within pore channels) are typically not aligned with the scales of
measurements and models (e.g., core-scale or above). The lack of clear understanding for how
variable inundation influences variation in biogeochemical processes and how these
relationships change with extent and grain of inundation can, therefore, lead to unreliable
predictions for the scaling of biogeochemical processes.



Understanding the biogeochemical influences of variable inundation across a broad range of
scales is important for informing a diverse suite of needs across models, decision makers, and
other interested parties. Our ability to inform these needs depends on our ability to rigorously
understand and predict influences of variable inundation across scales. This is a challenge as
variable inundation likely has direct, but unknown, influences over the scaling of biogeochemical
function. For example, cumulative metabolism in streams is predicted to increase faster than
their upstream drainage area for perennial stream networks (Wollheim et al. 2022). The
influence of variable inundation on biogeochemical processes cannot yet be accounted for in
such scaling theory. More generally, perturbations like variable inundation can drive systems
away from steady-state assumptions from which scaling relationships are derived (McCarthy et
al. 2019), therefore, we expect significant changes in scaling behavior across inundation
regimes. A research frontier is to quantify the direction, magnitude, and duration of changes in
scaling patterns in response to variable inundation and to modified variable inundation regimes
wrought by climate, land-use, and other environmental changes.

## Summary of Primary Methods used to Study VIEs

The multi-scale nature of VIE systems has led to experimental and observational studies that
span from point-scale lab-based characterization, to reach- or watershed-scale monitoring
networks, and to regional- and global-scale remote sensing. Point-scale measurements at the
smallest scales help reveal processes that underlie larger scale dynamics. For example, point
measures of water presence, water absence, and low flow detection within a watershed are
increasingly available with the development of small, inexpensive, and easily deployable
sensors, meters, and time-lapse cameras [e.g., (Soupir et al. 2009, Chapin et al. 2014, Costigan
et al. 2017, Zimmer et al. 2020)] (**Fig. 11**). While these measurements are easy to take and can
provide a long temporal dataset for little effort, they are not always detailed and require regular
calibrations.
A broad range of methods can be used to link the hydrologic dynamics to ecological and
biogeochemical responses. Standardized field surveys and biomolecular methods (e.g., isotopic
ratios, including compound specific analyses) are commonly used to study organismal,
population, and community ecology across multiple taxa [e.g., (Ode et al. 2016, Gates et al.
2020)] and can be standardized for both inundated and non-inundated states. There is
increasing use of crowdsourcing for biogeochemical characterization to consistently obtain
samples across diverse systems (von Schiller et al. 2019, Garayburu-Caruso et al. 2020).
Sample collection can be followed by a variety of laboratory measurements of properties (e.g.,
carbon content, redox potential and redox-active elements, microbial genetic potential, sediment
grain size) and processes, such as $CO_2$ production and methanogenesis related to variable
inundation. Point-scale measurements often operate at instantaneous to daily scales.
Conversely, larger scale measurements integrate across finer-scale processes to quantify
ecosystem dynamics and properties, but without necessarily revealing the governing the
processes. Spatially distributed monitoring networks using *in situ* sensors (e.g., the United
States Geological Survey, USGS, gage network) can connect event-scale responses across
hydrologically linked locations as well as reveal long-term trends [e.g., (Zipper et al. 2021)].
Long-term field manipulations are another complementary *in situ* technique that can reveal



mechanisms underlying system responses to changes in inundation state. There are numerous
configurations of such experiments that directly or indirectly impact inundation dynamics, such
as intentional inundation (Hopple et al. 2023), water exclusion (Kundel et al. 2018) and heating
(Hanson et al. 2017). Despite the plethora of data produced by such large scale projects, these
are expensive and require deep buy-in of researchers and landowners.
Remote sensing can complement *in situ* measurements to facilitate more spatially
continuous characterization of surface water dynamics and their impacts. There are different
types of remote sensing techniques that can capture different aspects of VIEs. For example, soil
surface saturation may be captured through passive microwave radiometer as well as C and L-
band radar backscatter, which can also penetrate through thin canopies, clouds, and through
the top few centimeters of the soil (Schumann and Moller 2015). Recent missions such as the
Surface Water and Ocean Topography (SWOT) mission provide increased capabilities for
monitoring changes in surface water over time with radar data (Biancamaria et al. 2016), while
NASA's forthcoming NISAR mission will allow for detection of inundation even under tree
canopy. Thermal infrared measurements can indirectly reveal saturation at very high
spatiotemporal resolutions, as well as evapotranspiration associated with water table depth, soil
moisture, and rooting depth (Fisher et al. 2020, Lalli et al. 2022). Long time series from
moderate resolution (~30 m) optical satellites can document multi-decadal open water trends
and seasonal regimes across the globe (Pekel et al. 2016), while some combinations of indices
have shown success in detecting mixed vegetation and inundation cover (Jones 2019).
Recently launched satellite constellations provide daily global imagery at < 4 m resolution,
enabling monitoring of more dynamic water bodies [e.g., Arctic lakes, (Cooley et al. 2017) and
forested wetlands (Hondula et al. 2021a)]. Finally, deep groundwater and changes in the total
water column storage are detectable through measurements of gravitational anomalies at very
high precision but low spatial resolution (Bloom et al. 2010, 2017, Richey et al. 2015, Pascolini-
Campbell et al. 2021).

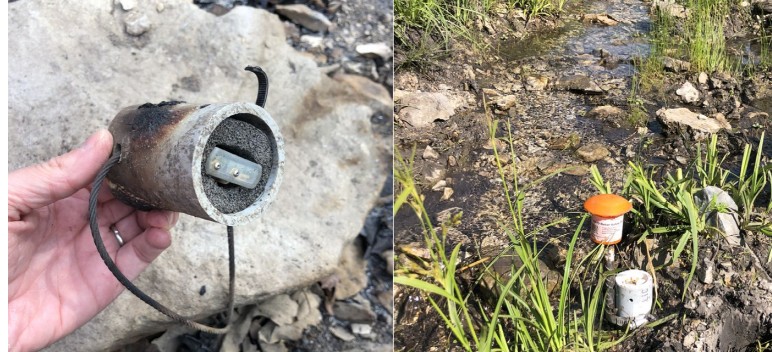

**Figure 11**. *Monitoring inundation regimes is increasingly possible via in situ sensors.*
*Stream Temperature, Intermittency, and Conductivity Sensors (STICs) (Chapin et al. 2014), one*
*of the types of increasingly available sensors to measure water presence/absence in an*
*inexpensive and easily deployable manner. These sensors can be used across all types of VIEs.*
*Credit: Amy Burgin.*



To advance predictive understanding requires integration of data with models. Process-
based models can be used to simulate hydrological and biogeochemical processes under dry
and wet conditions (Fatichi et al. 2016, Li et al. 2017). These models are often built upon mass
conservation principles, with ordinary or partial differential equations that describe coupled
hydrological, ecological, and biogeochemical processes. They rely on existing knowledge on
processes, including, for example, theories or empirical relationships between discharge and
water storage (Wittenberg 1999), biogeochemical reaction rate dependence on temperature and
water content (Davidson et al. 1998, Mahecha et al. 2010) and redox reactions (Borch et al.
2010). Among process-based models, there are spatially distributed models that couple surface
and subsurface flow dynamics explicitly (Kollet and Maxwell 2006, Coon et al. 2020). This class
of models has recently been extended to include reactive transport (Wu et al. 2021), which may
be considered as a set of tools to understand the biogeochemical effects of variable inundation
(Molins et al. 2022). However, spatial resolution and data requirements for the integrated
surface and subsurface models are high, which places practical limits on the spatial scales that
can be addressed. Semi- or fully-distributed models with coarse spatial resolution are able to
work at larger scales, but require theories or empirical relationships to represent processes and
impacts at subgrid-scales. Data-driven machine learning methods present new opportunities to
blend models with various levels of mechanistic representations into hybrid models (Reichstein
et al. 2019). Increases in the volume of observational data sets combined with advances in high
performance computing have triggered a shift towards machine learning applications for
capturing inundation dynamics. More recently, integration of physics-based models with
machine learning have improved the interpretability of machine learning methods and increased
their ability to model complex ecosystem processes (Sun et al. 2022). These hybrid approaches
have the potential to optimize the characterization and prediction of inundation dynamics by
incorporating the strengths of multiple models to achieve predictions with minimized uncertainty
and greater accuracy than either model alone.
Coordinated integration (Patel et al. 2023) between model development and data generation
is key to deepening our understanding of VIEs and increasing our ability to predict their future
ecosystem function and ecological integrity. More specifically, we promote iterating between
model-guided data generation and observation-informed model development. This iterative
cycle between models and 'experiments' (i.e., real-world data generation) has previously been
termed 'ModEx' (Atchley et al. 2015) and is similar to approaches used in 'ecological
forecasting' (Dietze et al. 2017, 2018). It also aligns generally with the scientific method based
on continuous iteration between conjectures (hypotheses / models) and refutation (falsification
of hypothesis using observations and data) to drive scientific discovery and knowledge growth
(Popper 2014). The ModEx approach often starts by using experimental or field data to
parameterize and calibrate models and/or generate new data based on known model input
needs. This can be expanded whereby models generate hypotheses via *in silico* experiments,
and field or lab studies can be designed to test those hypotheses. Models can also be used to
optimize the design of real-world experiments by indicating when, where, and what to measure
to provide the strongest hypothesis evaluation. In the context of VIEs, we expect ModEx to
touch scales ranging from molecular microbiology to landscape ecology to regional ecosystem
function to Earth system elemental cycles. Key to enabling this is the further development of
models and measurement techniques that can capture system states in both inundated and



non-inundated conditions. Techniques/models designed for specific kinds of ecosystems (e.g.,
perennial rivers) may be difficult to adapt. This emphasizes a need to do ModEx using models
and measurements intentionally designed to span inundated and non-inundated system states.
Across the continuum of ModEx, it is important to consider the scales at which models and
measurements operate, as discussed above. The issues around scale could, in part, be
addressed by Integrated Coordinated Open Networked (ICON) science principles (Goldman et
al. 2022). ICON is based on intentional design of research efforts to be Integrated across
disciplines and scales, Coordinated across research efforts via consistent methods, Open
throughout the research lifecycle, and Networked across stakeholders to understand collective
needs. We propose using ICON principles for *in situ* data generation and remote sensing, jointly
guided by model-generated predictions (i.e., ModEx). Embedding ICON throughout the research
life cycle can help to ensure that new data are at the right scale and can be used to link
disciplines (e.g., hydrology, biogeochemistry, and community ecology). This can also ensure
that data are interoperable across VIEs, are available to everyone and connected to deep
metadata, and are useful to a broad range of stakeholders with interests spanning different
types and locations of VIEs. The use of ICON in cross-VIE science could bridge existing data
across multiple spatial and temporal scales, and potentially bridge gaps among VIEs.

## Towards Cross-VIE Transferable Understanding

We propose that a key goal for VIE science is the development and open sharing of knowledge,
models, algorithms, and data that transcend individual system types. This can enhance our
capacity to predict and protect the future of VIE function and integrity. Knowledge that crosses
VIE systems will inherently span scales and levels of certainty from predictable, sub-daily
inundation regimes to rare extreme events; integrating perspectives of these dynamic systems
can aid in understanding and anticipating tipping points of physical, chemical, and biological
components across VIEs. Development of such knowledge should be done via ModEx
approaches coupled with ICON principles, which can generate models that can be used across
VIEs with different physical settings and hydrologic dynamics. We suggest this can be achieved
by taking a continuum approach based on key physical characteristics of VIEs (**Fig. 12**). While
the categorical approach in the above mini-review sections was used to emphasize the breadth
of VIE systems, we encourage research efforts to move beyond those artificial bins by invoking
this continuum approach. For example, a dynamic, unified classification model has been
proposed in wetlands, including a suite of temporally variable ecological and geomorphological
characteristics (Lisenby et al. 2019). This framework has improved the understanding of human
impacts on wetlands and led to more effective management (Wierzbicki et al. 2020, Mandishona
and Knight 2022).
The impacts of variable inundation depend on multiple characteristics of the inundation
regimes (e.g., return interval and duration) and factors that influence those regimes (e.g.,
subsurface permeability, topography, climate, and vegetation). Furthermore, there are dynamic
attributes that influence process rates (e.g., water residence time and hydrologic connectivity),
which can create additional feedback to the impacts of inundation variation. We hypothesize
that despite this complexity, cross-VIE science can make progress towards transferable
understanding by studying impacts of variable inundation across relatively simple physical



variables that can be easily measured. Two such variables are inundation return interval and
topographic slope (**Fig. 12**).
Inundation return interval represents a key component of the continuum of inundation
regimes and may be considered as a forcing factor. Topographic slope represents a key
component of the continuum of VIE characteristics that influence how VIE systems respond to
inundation-based forcing. In turn, these two variables can jointly influence nearly every physical,
chemical, and biological aspect of VIEs. We do not, however, imply that these two variables
capture all relevant processes. Other variables such as sediment/soil mineralogy and climate
also have influences over biogeochemistry and community ecology. Nonetheless, we propose
that significant progress can be made towards cross-VIE understanding of the controls over
biogeochemistry and community ecology by pursuing the continuum approach via inundation
return internal and topographic slope.
The continuum approach can be applied to questions representing science challenges that
span all VIEs, such as how greenhouse gas fluxes and biological diversity respond to variable
inundation (**Fig. 12**). Similarly, metabolism research has suggested using a continuum of flow
predictability and light availability to better unify river metabolism research (Bernhardt et al.
2022). In this approach there is no need to bin VIEs into discrete categories, many of which
have varying definitions and levels of overlap. Rather, we can observe and study continuous
response surfaces across multiple physical axes and identify patterns within this quantitative
space. In addition to generating transferable understanding, bringing all VIEs together via the
continuum approach could help raise awareness of VIE diversity, importance, vulnerabilities,
and how they may change in the future. This may, in turn, help address the fact that VIEs are
often overlooked in terms of conservation and monitoring efforts (Calhoun et al. 2017, Hill et al.
2018, Krabbenhoft et al. 2022, Zimmer et al. 2022). The continuum approach can also be used
to learn where, along environmental continuums, functional thresholds exist that could help with
categorizations important for policy and management (Richardson et al. 2022b).
Cross-VIE understanding of the drivers, patterns, and processes linking inundation to
system responses can greatly improve with increased collaboration and communication across
scientific fields and systems. Communities working in VIEs are scattered across different
societies and funding programs. Studying VIEs together via the continuum approach can bring
these science communities together. To this end, we encourage training and collaborations
focused on consistent data generation methods that may be adopted across the VIE community
and in pursuit of the continuum approach. In addition, disciplinary conferences could also
recognize VIE commonalities with special sessions to bring people together from across the VIE
continuum to discuss research needs.
Cross-VIE knowledge and models are needed to address human impacts to environments
across the globe. Humans both directly (i.e., dams, weirs, surface water and groundwater
abstraction, channelization, draining, invasive species introduction and spread, etc.) and
indirectly (i.e., climate change) alter VIEs. As climate change and other anthropogenic impacts
increasingly alter these already dynamic systems, it is imperative that knowledge and models
transcend VIEs. Future environmental change can alter the position of a given VIE within
environmental space, including what is depicted in **Fig. 12** (e.g., by changing the inundation
return interval). The ability to predict impacts of such environmental change can be facilitated by
mechanistic knowledge that is transferable across the environmental space occupied by VIEs.



We hypothesize that use of the continuum approach proposed here can be an effective
approach to achieving this mechanistic, transferable knowledge.

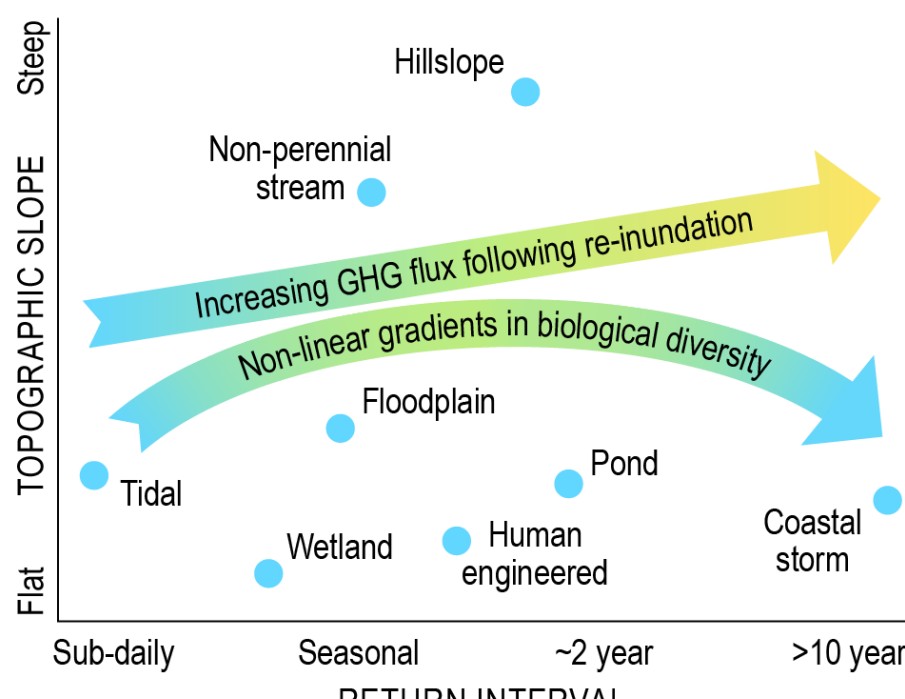

**Figure 12.** *We encourage a continuum perspective for VIE science whereby these*
*systems are studied across broad ranges of key controlling variables without regard for*
*what names may be attached to a given studied place and time.* *Two potential axes are*
*topographic slope and inundation return interval. Points represent approximate locations of*
*where each VIE type may lie. Each VIE type spans a range of slopes and inundation return*
*intervals, but we do not define these ranges as the continuum perspective is based on how*
*study systems fall across the environmental space represented here, rather than within specific*
*nomenclature. Two priority research directions are greenhouse gas (GHG) fluxes and biological*
*diversity, and the arrows represent possible hypotheses that could be evaluated with cross-VIE*
*studies. We propose that knowledge and models that are transferable across VIEs can be*
*achieved through evaluation of such hypotheses across broad environmental extents tied to key*
*environmental variables, such as slope and return interval. Credit: Nathan Johnson.*
**Acknowledgements**
Any use of trade, firm, or product names is for descriptive purposes only and does not imply
endorsement by the U.S. Government. The research described in this paper was supported by
the Earth & Biological Sciences Program Development Office at Pacific Northwest National



Laboratory, a multiprogram national laboratory operated by Battelle for the U.S. Department of
Energy. We thank Jon Chorover, Sarah Godsey, Jesus Gomez-Velez, Wei Huang, Roser
Matamala, Hyun Song and Kristen Underwood for contributions to the conceptual directions of
this manuscript. This manuscript was an outgrowth of the VIE Workshop and we greatly thank
the participants for their contributions.
**Competing Interests**: The authors declare that they have no conflict of interest.

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

A. M. De Girolamo. 2021. Analysing hydrological and sediment transport regime in two
Mediterranean intermittent rivers. CATENA 196:104865.
Fournier, R. J., G. De Mendoza, R. Sarremejane, and A. Ruhi. 2023. Isolation controls
reestablishment mechanisms and post-drying community structure in an intermittent
stream. Ecology 104:e3911.
Fredrickson, L., and T. S. Taylor. 1982. Management of seasonally flooded impoundments for
wildlife. Resource Publication 148, U.S. Fish and Wildlife Service.
Fryirs, K., and G. Brierley. 2022. Assemblages of geomorphic units: A building block approach
to analysis and interpretation of river character, behaviour, condition and recovery. Earth
Surface Processes and Landforms 47:92–108.
Gallant, A. 2015. The challenges of remote monitoring of wetlands. Remote Sensing 7:10938–

10950.



Garayburu-Caruso, V. A., R. E. Danczak, J. C. Stegen, L. Renteria, M. Mccall, A. E. Goldman,

R. K. Chu, J. Toyoda, C. T. Resch, J. M. Torgeson, J. Wells, S. Fansler, S. Kumar, and

E. B. Graham. 2020. Using community science to reveal the global chemogeography of

river metabolomes. Metabolites 10:518.

Gates, J. B., P. M. Chittaro, and K. B. Veggerby. 2020. Standard operating procedures for

measuring bulk stable isotope values of nitrogen and carbon in marine biota by isotope

ratio mass spectrometry (IRMS).

Gendreau, K. L., V. Buxton, C. E. Moore, and M. Mims. 2021. Temperature loggers capture

intraregional variation of inundation timing for intermittent ponds. preprint, Hydrology.

Gittman, R. K., F. J. Fodrie, A. M. Popowich, D. A. Keller, J. F. Bruno, C. A. Currin, C. H.

Peterson, and M. F. Piehler. 2015. Engineering away our natural defenses: an analysis

of shoreline hardening in the US. Frontiers in Ecology and the Environment 13:301–307.

Gleason, J. E., and R. C. Rooney. 2018. Pond permanence is a key determinant of aquatic

macroinvertebrate community structure in wetlands. Freshwater Biology 63:264–277.

Goldman, A. E., S. R. Emani, L. C. Pérez-Angel, J. A. Rodríguez-Ramos, and J. C. Stegen.

2022. Integrated, coordinated, open, and networked (ICON) science to advance the

geosciences: Introduction and synthesis of a special collection of commentary articles.

Earth and Space Science 9:e2021EA002099.

Goldman, A. E., E. B. Graham, A. R. Crump, D. W. Kennedy, E. B. Romero, C. G. Anderson, K.

1547         L. Dana, C. T. Resch, J. K. Fredrickson, and J. C. Stegen. 2017. Biogeochemical cycling

at the aquatic–terrestrial interface is linked to parafluvial hyporheic zone inundation

history. Biogeosciences 14:4229–4241.

Gómez-Gener, L., A. R. Siebers, M. I. Arce, S. Arnon, S. Bernal, R. Bolpagni, T. Datry, G.

Gionchetta, H.-P. Grossart, C. Mendoza-Lera, V. Pohl, U. Risse-Buhl, O. Shumilova, O.

Tzoraki, D. Von Schiller, A. Weigand, G. Weigelhofer, D. Zak, and A. Zoppini. 2021.

Towards an improved understanding of biogeochemical processes across surface-



groundwater interactions in intermittent rivers and ephemeral streams. Earth-Science
Reviews 220:103724.
González, E., A. A. Sher, E. Tabacchi, A. Masip, and M. Poulin. 2015. Restoration of riparian
vegetation: A global review of implementation and evaluation approaches in the
international, peer-reviewed literature. Journal of Environmental Management 158:85–

94.

Guimond, J. A., and H. A. Michael. 2021. Effects of marsh migration on flooding, saltwater
intrusion, and crop yield in coastal agricultural land subject to storm surge inundation.
Water Resources Research 57.
Hale, R. L., L. Turnbull, S. R. Earl, D. L. Childers, and N. B. Grimm. 2015. Stormwater
infrastructure controls runoff and dissolved material export from arid urban watersheds.
Ecosystems 18:62–75.
Hammond, J. C., M. Zimmer, M. Shanafield, K. Kaiser, S. E. Godsey, M. C. Mims, S. C. Zipper,
R. M. Burrows, S. K. Kampf, W. Dodds, C. N. Jones, C. A. Krabbenhoft, K. S. Boersma,
T. Datry, J. D. Olden, G. H. Allen, A. N. Price, K. Costigan, R. Hale, A. S. Ward, and D.
C. Allen. 2021. Spatial patterns and drivers of nonperennial flow regimes in the
contiguous United States. Geophysical Research Letters 48:e2020GL090794.
Hanson, P. J., J. S. Riggs, W. R. Nettles, J. R. Phillips, M. B. Krassovski, L. A. Hook, L. Gu, A.
D. Richardson, D. M. Aubrecht, D. M. Ricciuto, J. M. Warren, and C. Barbier. 2017.
Attaining whole-ecosystem warming using air and deep-soil heating methods with an
elevated $CO_2$ atmosphere. Biogeosciences 14:861–883.
Hayashi, M., G. Van Der Kamp, and D. O. Rosenberry. 2016. Hydrology of prairie wetlands:
Understanding the integrated surface-water and groundwater processes. Wetlands
36:237–254.



Herbert, E. R., J. Schubauer-Berigan, and C. B. Craft. 2018. Differential effects of chronic and
acute simulated seawater intrusion on tidal freshwater marsh carbon cycling.
Biogeochemistry 138:137–154.
Herndon, E. M., A. L. Dere, P. L. Sullivan, D. Norris, B. Reynolds, and S. L. Brantley. 2015.
Landscape heterogeneity drives contrasting concentration–discharge relationships in
shale headwater catchments. Hydrology and Earth System Sciences 19:3333–3347.
Herndon, E. M., G. Steinhoefel, A. L. D. Dere, and P. L. Sullivan. 2018. Perennial flow through
convergent hillslopes explains chemodynamic solute behavior in a shale headwater
catchment. Chemical Geology 493:413–425.
Herzon, I., and J. Helenius. 2008. Agricultural drainage ditches, their biological importance and
functioning. Biological Conservation 141:1171–1183.
Hill, M. J., H. M. Greaves, C. D. Sayer, C. Hassall, M. Milin, V. S. Milner, L. Marazzi, R. Hall, L.
R. Harper, I. Thornhill, R. Walton, J. Biggs, N. Ewald, A. Law, N. Willby, J. C. White, R.
A. Briers, K. L. Mathers, M. J. Jeffries, and P. J. Wood. 2021. Pond ecology and
conservation: research priorities and knowledge gaps. Ecosphere 12:e03853.
Hill, M. J., C. Hassall, B. Oertli, L. Fahrig, B. J. Robson, J. Biggs, M. J. Samways, N. Usio, N.
Takamura, J. Krishnaswamy, and P. J. Wood. 2018. New policy directions for global
pond conservation. Conservation Letters 11:e12447.
Hinkel, J., D. Lincke, A. T. Vafeidis, M. Perrette, R. J. Nicholls, R. S. J. Tol, B. Marzeion, X.
Fettweis, C. Ionescu, and A. Levermann. 2014. Coastal flood damage and adaptation
costs under 21st century sea-level rise. Proceedings of the National Academy of
Sciences 111:3292–3297.
Hinshaw, S. E., C. Tatariw, N. Flournoy, A. Kleinhuizen, C. Taylor, P. A. Sobecky, and B.
Mortazavi. 2017. Vegetation loss decreases salt marsh denitrification capacity:
Implications for marsh erosion. Environmental Science & Technology 51:8245–8253.



Hladyz, S., S. C. Watkins, K. L. Whitworth, and D. S. Baldwin. 2011. Flows and hypoxic
blackwater events in managed ephemeral river channels. Journal of Hydrology 401:117–

125.

Hofmeister, K. L., S. L. Eggert, B. J. Palik, D. Morley, E. Creighton, M. Rye, and R. K. Kolka.
2022. The identification, mapping, and management of seasonal ponds in forests of the
Great Lakes Region. Wetlands 42:9.
Hondula, K. L., B. DeVries, C. N. Jones, and M. A. Palmer. 2021a. Effects of Using High
Resolution Satellite-Based Inundation Time Series to Estimate Methane Fluxes From
Forested Wetlands. Geophysical Research Letters 48:e2021GL092556.
Hondula, K. L., C. N. Jones, and M. A. Palmer. 2021b. Effects of seasonal inundation on
methane fluxes from forested freshwater wetlands. Environmental Research Letters

16:084016.

Hooley-Underwood, Z. E., S. B. Stevens, N. R. Salinas, and K. G. Thompson. 2019. An
intermittent stream supports extensive spawning of large-river native fishes.
Transactions of the American Fisheries Society 148:426–441.
Hopple, A. M., K. O. Doro, V. L. Bailey, B. Bond-Lamberty, N. McDowell, K. A. Morris, A. Myers-
Pigg, S. C. Pennington, P. Regier, R. Rich, A. Sengupta, R. Smith, J. Stegen, N. D.
Ward, S. C. Woodard, and J. P. Megonigal. 2023. Attaining freshwater and estuarine-
water soil saturation in an ecosystem-scale coastal flooding experiment. Environmental
Monitoring and Assessment 195:425.
Hopple, A. M., S. C. Pennington, J. P. Megonigal, V. Bailey, and B. Bond-Lamberty. 2022.
Disturbance legacies regulate coastal forest soil stability to changing salinity and
inundation: A soil transplant experiment. Soil Biology and Biochemistry 169:108675.
Houser, C., and S. Hamilton. 2009. Sensitivity of post-hurricane beach and dune recovery to
event frequency. Earth Surface Processes and Landforms 34:613–628.



Huang, C., C. Gascuel-Odoux, and S. Cros-Cayot. 2002. Hillslope topographic and hydrologic

effects on overland flow and erosion. CATENA 46:177–188.

Huang, W., K. Wang, C. Ye, W. C. Hockaday, G. Wang, and S. J. Hall. 2021. High carbon

losses from oxygen-limited soils challenge biogeochemical theory and model

assumptions. Global Change Biology 27:6166–6180.

Ivory, S. J., M. M. McGlue, S. Spera, A. Silva, and I. Bergier. 2019. Vegetation, rainfall, and

pulsing hydrology in the Pantanal, the world's largest tropical wetland. Environmental

Research Letters 14:124017.

Jarecke, K. M., T. D. Loecke, and A. J. Burgin. 2016. Coupled soil oxygen and greenhouse gas

dynamics under variable hydrology. Soil Biology and Biochemistry 95:164–172.

Jeffries, M. 2008. The spatial and temporal heterogeneity of macrophyte communities in thirty

small, temporary ponds over a period of ten years. Ecography 31:765–775.

Jones, C. N., G. R. Evenson, D. L. McLaughlin, M. K. Vanderhoof, M. W. Lang, G. W. McCarty,

H. E. Golden, C. R. Lane, and L. C. Alexander. 2018. Estimating restorable wetland

water storage at landscape scales. Hydrological Processes 32:305–313.

Jones, J. 2019. Improved automated detection of subpixel-scale inundation: Revised dynamic

surface water extent (DSWE) partial surface water tests. Remote Sensing 11:374.

Junk, W., P. Bayley, and R. Sparks. 1989. The flood pulse concept in river-floodplain systems.

Page Can. Spec. Public Fish. Aquat. Sci.

Kaushal, S. S., J. E. Reimer, P. M. Mayer, R. R. Shatkay, C. M. Maas, W. D. Nguyen, W. L.

Boger, A. M. Yaculak, T. R. Doody, M. J. Pennino, N. W. Bailey, J. G. Galella, A.

Weingrad, D. C. Collison, K. L. Wood, S. Haq, T. A. Newcomer-Johnson, S. Duan, and

1650       K. T. Belt. 2022. Freshwater salinization syndrome alters retention and release of

chemical cocktails along flowpaths: From stormwater management to urban

streams. Freshwater Science 41:420–441.





Kirkby, M., L. Bracken, and S. Reaney. 2002. The influence of land use, soils and topography

on the delivery of hillslope runoff to channels in SE Spain. Earth Surface Processes and

Landforms 27:1459–1473.

Kirwan, M. L., and K. B. Gedan. 2019. Sea-level driven land conversion and the formation of

ghost forests. Nature Climate Change 9:450–457.

Kneitel, J. M. 2014. Inundation timing, more than duration, affects the community structure of

California vernal pool mesocosms. Hydrobiologia 732:71–83.

Kollet, S. J., and R. M. Maxwell. 2006. Integrated surface–groundwater flow modeling: A free-

surface overland flow boundary condition in a parallel groundwater flow model.

Advances in Water Resources 29:945–958.

Konapala, G., A. K. Mishra, Y. Wada, and M. E. Mann. 2020. Climate change will affect global

water availability through compounding changes in seasonal precipitation and

evaporation. Nature Communications 11:3044.

Koschorreck, M., A. S. Downing, J. Hejzlar, R. Marcé, A. Laas, W. G. Arndt, P. S. Keller, A. J. P.

Smolders, G. van Dijk, and S. Kosten. 2020. Hidden treasures: Human-made aquatic

ecosystems harbour unexplored opportunities. Ambio 49:531–540.

Krabbenhoft, C. A., G. H. Allen, P. Lin, S. E. Godsey, D. C. Allen, R. M. Burrows, A. G.

DelVecchia, K. M. Fritz, M. Shanafield, A. J. Burgin, M. A. Zimmer, T. Datry, W. K.

Dodds, C. N. Jones, M. C. Mims, C. Franklin, J. C. Hammond, S. Zipper, A. S. Ward, K.

H. Costigan, H. E. Beck, and J. D. Olden. 2022. Assessing placement bias of the global

river gauge network. Nature Sustainability 5:586–592.

Kundel, D., S. Meyer, H. Birkhofer, A. Fliessbach, P. Mäder, S. Scheu, M. van Kleunen, and K.

Birkhofer. 2018. Design and manual to construct rainout-shelters for climate shange

experiments in agroecosystems. Frontiers in Environmental Science 6.

Ladau, J., and E. A. Eloe-Fadrosh. 2019. Spatial, temporal, and phylogenetic scales of microbial

ecology. Trends in Microbiology 27:662–669.



Lalli, K., S. Soenen, J. B. Fisher, J. McGlinchy, T. Kleynhans, R. Eon, and L. M. Moreau. 2022.

VanZyl-1: demonstrating SmallSat measurement capabilities for land surface

temperature and evapotranspiration. Page 8 *in* C. D. Norton and S. R. Babu, editors.

CubeSats and SmallSats for Remote Sensing VI. SPIE, San Diego, United States.

Lane, C. R., and E. D'Amico. 2016. Identification of putative geographically isolated wetlands of

the conterminous United States. JAWRA Journal of the American Water Resources

Association 52:705–722.

Lane, K., K. Charles-Guzman, K. Wheeler, Z. Abid, N. Graber, and T. Matte. 2013. Health

effects of coastal storms and flooding in urban areas: A review and vulnerability

assessment. Journal of Environmental and Public Health 2013:1–13.

Laronne, J. B., and L. Reid. 1993. Very high rates of bedload sediment transport by ephemeral

desert rivers. Nature 366:148–150.

Larsen, L., N. Aumen, C. Bernhardt, V. Engel, T. Givnish, S. Hagerthey, J. Harvey, L. Leonard,

P. McCormick, C. Mcvoy, G. Noe, M. Nungesser, K. Rutchey, F. Sklar, T. Troxler, J.

Volin, and D. Willard. 2011. Recent and historic drivers of landscape change in the

Everglades Ridge, slough, and tree island mosaic. Critical Reviews in Environmental

Science and Technology 41:344–381.

Lei, Y., C. Liu, L. Zhang, and S. Luo. 2016. How smallholder farmers adapt to agricultural

drought in a changing climate: A case study in southern China. Land Use Policy 55:300–

308.

Li, L., K. Maher, A. Navarre-Sitchler, J. Druhan, C. Meile, C. Lawrence, J. Moore, J. Perdrial, P.

Sullivan, A. Thompson, L. Jin, E. W. Bolton, S. L. Brantley, W. E. Dietrich, K. U. Mayer,

C. I. Steefel, A. Valocchi, J. Zachara, B. Kocar, J. Mcintosh, B. M. Tutolo, M. Kumar, E.

Sonnenthal, C. Bao, and J. Beisman. 2017. Expanding the role of reactive transport

models in critical zone processes. Earth-Science Reviews 165:280–301.



Li, L., P. L. Sullivan, P. Benettin, O. A. Cirpka, K. Bishop, S. L. Brantley, J. L. A. Knapp, I.

Meerveld, A. Rinaldo, J. Seibert, H. Wen, and J. W. Kirchner. 2021. Toward catchment

hydro-biogeochemical theories. WIREs Water 8.

Li, S., G. Wang, C. Zhu, J. Lu, W. Ullah, D. F. T. Hagan, G. Kattel, and J. Peng. 2022a.

Attribution of global evapotranspiration trends based on the Budyko framework.

Hydrology and Earth System Sciences 26:3691–3707.

Li, Z., S. Gao, M. Chen, J. J. Gourley, and Y. Hong. 2022b. Spatiotemporal characteristics of

US floods: Current status and forecast under a future warmer climate. Earth's Future

10:e2022EF002700.

Liberato, M. L. R., J. G. Pinto, R. M. Trigo, P. Ludwig, P. Ordóñez, D. Yuen, and I. F. Trigo.

2013. Explosive development of winter storm Xynthia over the subtropical North Atlantic

Ocean. Natural Hazards and Earth System Sciences 13:2239–2251.

Lisenby, P. E., S. Tooth, and T. J. Ralph. 2019. Product vs. process? The role of

geomorphology in wetland characterization. Science of The Total Environment 663:980–

991.

Londe, D. W., D. Dvorett, C. A. Davis, S. R. Loss, and E. P. Robertson. 2022a. Inundation of

depressional wetlands declines under a changing climate. Climatic Change 172:27.

Londe, D. W., D. Dvorett, C. A. Davis, S. R. Loss, and E. P. Robertson. 2022b. Inundation of

depressional wetlands declines under a changing climate. Climatic Change 172:27.

Lovelock, C. E., and R. Reef. 2020. Variable impacts of climate change on blue carbon. One

Earth 3:195–211.

Lugo, A. E. 2008. Visible and invisible effects of hurricanes on forest ecosystems: an

international review. Austral Ecology 33:368–398.

Luijendijk, A., G. Hagenaars, R. Ranasinghe, F. Baart, G. Donchyts, and S. Aarninkhof. 2018.

The state of the world's beaches. Scientific Reports 8:6641.



Mahecha, M. D., M. Reichstein, N. Carvalhais, G. Lasslop, H. Lange, S. I. Seneviratne, R.

Vargas, C. Ammann, M. A. Arain, A. Cescatti, I. A. Janssens, M. Migliavacca, L.

Montagnani, and A. D. Richardson. 2010. Global convergence in the temperature

sensitivity of respiration at ecosystem level. Science 329:838–840.

Mandishona, E., and J. Knight. 2022. Inland wetlands in Africa: A review of their typologies and

ecosystem services. Progress in Physical Geography: Earth and Environment 46:547–

565.

Marton, J. M., I. F. Creed, D. B. Lewis, C. R. Lane, N. B. Basu, M. J. Cohen, and C. B. Craft.

2015. Geographically isolated wetlands are important biogeochemical reactors on the

landscape. BioScience 65:408–418.

Matthews, J. 2010. Anthropogenic climate change impacts on ponds: a thermal mass

perspective. BioRisk 5:193–209.

Maul, G. A., and I. W. Duedall. 2019. Demography of coastal populations. Pages 692–700 *in* C.

1742       W. Finkl and C. Makowski, editors. Encyclopedia of Coastal Science. Springer

International Publishing, Cham.

McCarthy, J. K., J. M. Dwyer, and K. Mokany. 2019. A regional-scale assessment of using

metabolic scaling theory to predict ecosystem properties. Proceedings of the Royal

Society B: Biological Sciences 286:20192221.

McClain, M. E., E. W. Boyer, C. L. Dent, S. E. Gergel, N. B. Grimm, P. M. Groffman, S. C. Hart,

1748       J. W. Harvey, C. A. Johnston, E. Mayorga, W. H. McDowell, and G. Pinay. 2003.

Biogeochemical hot spots and hot moments at the interface of terrestrial and aquatic

ecosystems. Ecosystems 6:301–312.

McDaniel, P. A., M. P. Regan, E. Brooks, J. Boll, S. Barndt, A. Falen, S. K. Young, and J. E.

Hammel. 2008. Linking fragipans, perched water tables, and catchment-scale

hydrological processes. CATENA 73:166–173.



McDonnell, J. J. 2009. Hewlett, J.D. and Hibbert, A.R. 1967: Factors affecting the response of

small watersheds to precipitation in humid areas. In Sopper, W.E. and Lull, H.W.,

editors, Forest hydrology, New York: Pergamon Press, 275—90. Progress in Physical

Geography: Earth and Environment 33:288–293.

McDowell, N. G., M. Ball, B. Bond-Lamberty, M. L. Kirwan, K. W. Krauss, J. P. Megonigal, M.

Mencuccini, N. D. Ward, M. N. Weintraub, and V. Bailey. 2022. Processes and

mechanisms of coastal woody-plant mortality. Global Change Biology 28:5881–5900.

Mcleod, E., G. L. Chmura, S. Bouillon, R. Salm, M. Björk, C. M. Duarte, C. E. Lovelock, W. H.

Schlesinger, and B. R. Silliman. 2011. A blueprint for blue carbon: toward an improved

understanding of the role of vegetated coastal habitats in sequestering CO2. Frontiers in

Ecology and the Environment 9:552–560.

McVicar, T. R., T. G. Van Niel, L. Li, M. F. Hutchinson, X. Mu, and Z. Liu. 2007. Spatially

distributing monthly reference evapotranspiration and pan evaporation considering

topographic influences. Journal of Hydrology 338:196–220.

Melton, J. R., R. Wania, E. L. Hodson, B. Poulter, B. Ringeval, R. Spahni, T. Bohn, C. A. Avis,

D. J. Beerling, G. Chen, A. V. Eliseev, S. N. Denisov, P. O. Hopcroft, D. P. Lettenmaier,

1770         W. J. Riley, J. S. Singarayer, Z. M. Subin, H. Tian, S. Zürcher, V. Brovkin, P. M. Van

Bodegom, T. Kleinen, Z. C. Yu, and J. O. Kaplan. 2013. Present state of global wetland

extent and wetland methane modelling: conclusions from a model inter-comparison

project (WETCHIMP). Biogeosciences 10:753–788.

Merritt, D. M., and E. E. Wohl. 2002. Processes governing hydrochory along rivers: Hydraulics,

hydrology and dispersal phenology. Ecological Applications 12:1071–1087.

Mertes, L. A. K. 2011. Inland flood hazards: Human, riparian, and aquatic communities. Pages

145–166 Inundation hydrology. Cambridge University Press, Cambridge, UK.



Messager, M. L., B. Lehner, C. Cockburn, N. Lamouroux, H. Pella, T. Snelder, K. Tockner, T.

Trautmann, C. Watt, and T. Datry. 2021. Global prevalence of non-perennial rivers and

streams. Nature 594:391–397.

Mikac, S., A. Zmegac, T. Domagoj, P. Vinko, M. Orsanic, and I. Anic. 2018. Drought-induced

shift in tree response to climate in floodplain forests of Southeastern Europe. Scientific

Reports 8:16495.

Molins, S., D. Svyatsky, Z. Xu, E. T. Coon, and J. D. Moulton. 2022. A multicomponent reactive

transport model for integrated surface-subsurface hydrology problems. Water Resources

Research 58:e2022WR032074.

Moomaw, W. R., G. L. Chmura, G. T. Davies, C. M. Finlayson, B. A. Middleton, S. M. Natali, J.

E. Perry, N. Roulet, and A. E. Sutton-Grier. 2018. Wetlands In a changing climate:

Science, policy and management. Wetlands 38:183–205.

Morrissey, E. M., and R. B. Franklin. 2015. Evolutionary history influences the salinity

preference of bacterial taxa in wetland soils. Frontiers in Microbiology 6.

Murray, N. J., P. Bunting, R. F. Canto, L. Hilarides, E. V. Kennedy, R. M. Lucas, M. B. Lyons, A.

Navarro, C. M. Roelfsema, A. Rosenqvist, M. D. Spalding, M. Toor, and T. A.

Worthington. 2022a. coastTrain: A global feference library for coastal ecosystems.

Remote Sensing 14:5766.

Murray, N. J., T. A. Worthington, P. Bunting, S. Duce, V. Hagger, C. E. Lovelock, R. Lucas, M. I.

Saunders, M. Sheaves, M. Spalding, N. J. Waltham, and M. B. Lyons. 2022b. High-

resolution mapping of losses and gains of Earth's tidal wetlands. Science 376:744–749.

Nanson, G., and J. Croke. 1992. A genetic classification of floodplains. Faculty of Science,

Medicine and Health - Papers: part A:459–486.

Nelson, T. M., C. Streten, K. S. Gibb, and A. A. Chariton. 2015. Saltwater intrusion history

shapes the response of bacterial communities upon rehydration. Science of The Total

Environment 502:143–148.



Neubauer, S. C., and J. P. Megonigal. 2015. Moving beyond global warming potentials to

quantify the climatic role of ecosystems. Ecosystems 18:1000–1013.

Ode, P. R., A. E. Fetscher, and L. B. Busse. 2016. Standard operating procedures for the

collection of field data for bioassessments of California wadeable streams: Benthic

macroinvertebrates, algae, and physical habitat. Page 80. California State Water

Resources Control Board Surface Water Ambient Monitoring Program (SWAMP)

Bioassessment SOP 004.

O'Mara, K., J. M. Olley, B. Fry, and M. Burford. 2019. Catchment soils supply ammonium to the

coastal zone - Flood impacts on nutrient flux in estuaries. Science of The Total

Environment 654:583–592.

O'Meara, T. A., J. R. Hillman, and S. F. Thrush. 2017. Rising tides, cumulative impacts and

cascading changes to estuarine ecosystem functions. Scientific Reports 7:10218.

Palmer, M. A., and K. L. Hondula. 2014. Restoration as Mitigation: Analysis of Stream Mitigation

for Coal Mining Impacts in Southern Appalachia. Environmental Science & Technology

48:10552–10560.

Palmer, M. A., K. L. Hondula, and B. J. Koch. 2014. Ecological restoration of streams and rivers:

Shifting strategies and shifting goals. Annual Review of Ecology, Evolution, and

Systematics 45:247–269.

Palta, M. M., J. G. Ehrenfeld, and P. M. Groffman. 2014. "Hotspots" and "Hot Moments" of

Denitrification in Urban Brownfield Wetlands. Ecosystems 17:1121–1137.

Palta, M. M., N. B. Grimm, and P. M. Groffman. 2017. "Accidental" urban wetlands: Ecosystem

functions in unexpected places. Frontiers in Ecology and the Environment 15:248–256.

Pan, J., Y. Liu, X. Zhong, R. M. Lampayan, G. R. Singleton, N. Huang, K. Liang, B. Peng, and

1827        K. Tian. 2017. Grain yield, water productivity and nitrogen use efficiency of rice under

different water management and fertilizer-N inputs in South China. Agricultural Water

Management 184:191–200.



Pascolini-Campbell, M., J. B. Fisher, and J. T. Reager. 2021. GRACE-FO and ECOSTRESS

synergies constrain fine-scale impacts on the water balance. Geophysical Research

Letters 48.

Patel, K. F., K. A. Rod, J. Zheng, P. J. Regier, F. Machado-Silva, B. Bond-Lamberty, X. Chen,

D. Day, K. O. Doro, M. Kaufman, M. Kovach, N. McDowell, S. A. McKever, P. J.

Megonigal, C. G. Norris, T. O'Meara, R. Rich, P. Thornton, K. M. Kemner, N. D. Ward,

1836       M. N. Weintraub, and V. L. Bailey. 2023. Time to anoxia: Observations and predictions

of oxygen drawdown following coastal flood events.

Patel, K. F., C. Tatariw, J. D. MacRae, T. Ohno, S. J. Nelson, and I. J. Fernandez. 2020.

Snowmelt periods as hot moments for soil N dynamics: a case study in Maine, USA.

Environmental Monitoring and Assessment 192:777.

Patel, N., S. Gahlaud, A. Saxena, B. Thakur, N. Bharti, A. Dabhi, R. Bhushan, and R. Agnihotri.

2022. Revised chronology and stable isotopic (carbon and nitrogen) characterization of

Lahuradewa lake sediment (Ganga-plain, India): Insights into biogeochemistry leading to

peat formation in the lake. Journal of the Palaeontological Society of India Volume

67(1):113–125.

Peacock, M., J. Audet, D. Bastviken, M. N. Futter, V. Gauci, A. Grinham, J. A. Harrison, M. S.

Kent, S. Kosten, C. E. Lovelock, A. J. Veraart, and C. D. Evans. 2021. Global

importance of methane emissions from drainage ditches and canals. Environmental

Research Letters 16.

Pedersen, O., M. Sauter, T. D. Colmer, and M. Nakazono. 2021. Regulation of root adaptive

anatomical and morphological traits during low soil oxygen. New Phytologist 229:42–49.

Pekel, J.-F., A. Cottam, N. Gorelick, and A. S. Belward. 2016. High-resolution mapping of global

surface water and its long-term changes. Nature 540:418–422.



Peng, S., X. Lin, R. L. Thompson, Y. Xi, G. Liu, D. Hauglustaine, X. Lan, B. Poulter, M. Ramonet, M. Saunois, Y. Yin, Z. Zhang, B. Zheng, and P. Ciais. 2022. Wetland emission and atmospheric sink changes explain methane growth in 2020. Nature 612:477–482.

Peruccacci, S., M. T. Brunetti, S. L. Gariano, M. Melillo, M. Rossi, and F. Guzzetti. 2017. Rainfall thresholds for possible landslide occurrence in Italy. Geomorphology 290:39–57.

Pezeshki, S. R., and R. D. DeLaune. 2012. Soil oxidation-reduction in wetlands and Its impact on plant functioning. Biology 1:196–221.

Pickering, M. D., K. J. Horsburgh, J. R. Blundell, J. J.-M. Hirschi, R. J. Nicholls, M. Verlaan, and N. C. Wells. 2017. The impact of future sea-level rise on the global tides. Continental Shelf Research 142:50–68.

Plum, N. 2005. Terrestrial invertebrates in flooded grassland: A literature review. Wetlands 25:721–737.

Pool, S., F. Francés, A. Garcia-Prats, M. Pulido-Velazquez, C. Sanchis-Ibor, M. Schirmer, H. Yang, and J. Jiménez-Martínez. 2021. From flood to drip irrigation under climate change: Impacts on evapotranspiration and groundwater recharge in the mediterranean region of Valencia (Spain). Earth's Future 9:e2020EF001859.

Popper, K. R. 2014. Conjectures and refutations: the growth of scientific knowledge. Repr. Routledge, London.

Price, A. N., C. N. Jones, J. C. Hammond, M. A. Zimmer, and S. C. Zipper. 2021. The drying regimes of non-perennial rivers and streams. Geophysical Research Letters 48:e2021GL093298.

Quinn, J. D., P. M. Reed, M. Giuliani, A. Castelletti, J. W. Oyler, and R. E. Nicholas. 2018. Exploring how changing monsoonal dynamics and human pressures challenge multireservoir management for flood protection, hydropower production, and agricultural water supply. Water Resources Research 54:4638–4662.



Rameshwaran, P., V. A. Bell, H. N. Davies, and A. L. Kay. 2021. How might climate change

affect river flows across West Africa? Climatic Change 169:21.

Rasmussen, T. C., J. B. Deemy, and S. L. Long. 2016. Wetland Hydrology. Pages 1–16 *in* C. M.

Finlayson, M. Everard, K. Irvine, R. J. McInnes, B. A. Middleton, A. A. Van Dam, and N.

C. Davidson, editors. The Wetland Book. Springer Netherlands, Dordrecht.

Regier, P., N. D. Ward, J. Indivero, C. Wiese Moore, M. Norwood, and A. Myers-Pigg. 2021.

Biogeochemical control points of connectivity between a tidal creek and its floodplain.

Limnology and Oceanography Letters 6:134–142.

Reichstein, M., G. Camps-Valls, B. Stevens, M. Jung, J. Denzler, N. Carvalhais, and Prabhat.

2019. Deep learning and process understanding for data-driven Earth system science.

Nature 566:195–204.

Reis, V., V. Hermoso, S. K. Hamilton, D. Ward, E. Fluet-Chouinard, B. Lehner, and S. Linke.

2017. A global assessment of inland wetland conservation status. BioScience 67:523–

533.

Reisinger, A. J., P. M. Groffman, and E. J. Rosi-Marshall. 2016. Nitrogen cycling process rates

across urban ecosystems. FEMS Microbiology Ecology 92:fiw198.

Renwick, W., R. Sleezer, R. Buddemeier, and S. Smith. 2006. Small artificial ponds in the

United States: Impacts on sedimentation and carbon budget. Pages 738–744

Proceedings of the Eighth Federal Interagency Sedimentation Conference.

Resetarits, W. J. 1996. Oviposition site choice and life history evolution. American Zoologist

36:205–215.

Reverey, F., L. Ganzert, G. Lischeid, A. Ulrich, K. Premke, and H.-P. Grossart. 2018. Dry-wet

cycles of kettle hole sediments leave a microbial and biogeochemical legacy. Science of

The Total Environment 627:985–996.

Richardson, D. C., M. A. Holgerson, M. J. Farragher, K. K. Hoffman, K. B. S. King, M. B.

Alfonso, M. R. Andersen, K. S. Cheruveil, K. A. Coleman, M. J. Farruggia, R. L.



Fernandez, K. L. Hondula, G. A. López Moreira Mazacotte, K. Paul, B. L. Peierls, J. S.

Rabaey, S. Sadro, M. L. Sánchez, R. L. Smyth, and J. N. Sweetman. 2022a. A

functional definition to distinguish ponds from lakes and wetlands. Scientific Reports

12:10472.

Richardson, D. C., M. A. Holgerson, M. J. Farragher, K. K. Hoffman, K. B. S. King, M. B.

Alfonso, M. R. Andersen, K. S. Cheruveil, K. A. Coleman, M. J. Farruggia, R. L.

Fernandez, K. L. Hondula, G. A. López Moreira Mazacotte, K. Paul, B. L. Peierls, J. S.

Rabaey, S. Sadro, M. L. Sánchez, R. L. Smyth, and J. N. Sweetman. 2022b. A

functional definition to distinguish ponds from lakes and wetlands. Scientific Reports

12:10472.

Richey, A. S., B. F. Thomas, M. Lo, J. T. Reager, J. S. Famiglietti, K. Voss, S. Swenson, and M.

Rodell. 2015. Quantifying renewable groundwater stress with GRACE. Water Resources

Research 51:5217–5238.

Ripley, B. J., and M. A. Simovich. 2009. Species richness on islands in time: Variation in

ephemeral pond crustacean communities in relation to habitat duration and size.

Hydrobiologia 617:181–196.

Robinson, C. T., K. Tockner, and J. V. Ward. 2002. The fauna of dynamic riverine landscapes:

Fauna of riverine landscapes. Freshwater Biology 47:661–677.

Rosado, J., M. Morais, and K. Tockner. 2015. Mass dispersal of terrestrial organisms during first

flush events in a temporary stream: Mass dispersal of terrestrial organisms. River

Research and Applications 31:912–917.

Rosentreter, J. A., A. V. Borges, B. R. Deemer, M. A. Holgerson, S. Liu, C. Song, J. Melack, P.

1927        A. Raymond, C. M. Duarte, G. H. Allen, D. Olefeldt, B. Poulter, T. I. Battin, and B. D.

Eyre. 2021. Half of global methane emissions come from highly variable aquatic

ecosystem sources. Nature Geoscience 14:225–230.



Ruel, J. J., and M. P. Ayres. 1999. Jensen's inequality predicts effects of environmental

variation. Trends in Ecology & Evolution 14:361–366.

Rullens, V., S. Mangan, F. Stephenson, D. E. Clark, R. H. Bulmer, A. Berthelsen, J. Crawshaw,

R. V. Gladstone-Gallagher, S. Thomas, J. I. Ellis, and C. A. Pilditch. 2022.

Understanding the consequences of sea level rise: the ecological implications of losing

intertidal habitat. New Zealand Journal of Marine and Freshwater Research 56:353–370.

Saadat, S., J. Frankenberger, L. Bowling, and S. Ale. 2020. Evaluation of surface ponding and

runoff generation in a seasonally frozen drained agricultural field. Journal of Hydrology

588:124985.

Saltarelli, W. A., D. G. F. Cunha, A. Freixa, N. Perujo, J. C. López-Doval, V. Acuña, and S.

Sabater. 2022. Nutrient stream attenuation is altered by the duration and frequency of

flow intermittency. Ecohydrology 15:e2351.

Sarremejane, R., H. Mykrä, N. Bonada, J. Aroviita, and T. Muotka. 2017. Habitat connectivity

and dispersal ability drive the assembly mechanisms of macroinvertebrate communities

in river networks. Freshwater Biology 62.

Sarremejane, R., R. Stubbington, J. England, C. E. M. Sefton, M. Eastman, S. Parry, and A.

Ruhi. 2021. Drought effects on invertebrate metapopulation dynamics and quasi-

extinction risk in an intermittent river network. Global Change Biology 27:4024–4039.

Schaffer-Smith, D., S. W. Myint, R. L. Muenich, D. Tong, and J. E. DeMeester. 2020. Repeated

hurricanes reveal risks and opportunities for social-ecological resilience to flooding and

water quality problems. Environmental Science & Technology 54:7194–7204.

von Schiller, D., T. Datry, R. Corti, A. Foulquier, K. Tockner, R. Marcé, G. García-Baquero, I.

Odriozola, B. Obrador, A. Elosegi, C. Mendoza-Lera, M. O. Gessner, R. Stubbington, R.

Albariño, D. C. Allen, F. Altermatt, M. I. Arce, S. Arnon, D. Banas, A. Banegas-Medina,

E. Beller, M. L. Blanchette, J. F. Blanco-Libreros, J. Blessing, I. G. Boëchat, K. S.

Boersma, M. T. Bogan, N. Bonada, N. R. Bond, K. Brintrup, A. Bruder, R. M. Burrows, T.



Cancellario, S. M. Carlson, S. Cauvy-Fraunié, N. Cid, M. Danger, B. de Freitas Terra, A.

Dehedin, A. M. De Girolamo, R. del Campo, V. Díaz-Villanueva, C. P. Duerdoth, F. Dyer,

E. Faye, C. Febria, R. Figueroa, B. Four, S. Gafny, R. Gómez, L. Gómez-Gener, M. a. S.

Graça, S. Guareschi, B. Gücker, F. Hoppeler, J. L. Hwan, S. Kubheka, A. Laini, S. D.

Langhans, C. Leigh, C. J. Little, S. Lorenz, J. Marshall, E. J. Martín, A. McIntosh, E. I.

Meyer, M. Miliša, M. C. Mlambo, M. Moleón, M. Morais, P. Negus, D. Niyogi, A.

Papatheodoulou, I. Pardo, P. Pařil, V. Pešić, C. Piscart, M. Polášek, P. Rodríguez-

Lozano, R. J. Rolls, M. M. Sánchez-Montoya, A. Savić, O. Shumilova, A. Steward, A.

Taleb, A. Uzan, R. Vander Vorste, N. Waltham, C. Woelfle-Erskine, D. Zak, C. Zarfl, and

1965        A. Zoppini. 2019. Sediment respiration pulses in intermittent rivers and ephemeral

streams. Global Biogeochemical Cycles 33:1251–1263.

Schimel, J. P. 2018. Life in dry soils: Effects of drought on soil microbial communities and

processes. Annual Review of Ecology, Evolution, and Systematics 49:409–432.

Schlesinger, W. H., and E. S. Bernhardt. 2020. The atmosphere. Pages 51–97

Biogeochemistry. Elsevier.

Schumann, G. J.-P., and D. K. Moller. 2015. Microwave remote sensing of flood inundation.

Physics and Chemistry of the Earth, Parts A/B/C 83–84:84–95.

Scientific Investigations Report. 2015. . Scientific Investigations Report.
Secretariat, R. 2016. An Introduction to the Convention on Wetlands (previously The Ramsar

Convention Manual). 7th edition.

Semeniuk, C. A., and V. Semeniuk. 1995. A geomorphic approach to global classification for

inland wetlands. Pages 103–124 Advances in Vegetation Science.

Semeniuk, C., and V. Semeniuk. 2011. A comprehensive classification of inland wetlands of

Western Australia using the geomorphic-hydrologic approach. Journal of the Royal

Society of Western Australia 94:449–464.



Shi, X., P. E. Thornton, D. M. Ricciuto, P. J. Hanson, J. Mao, S. D. Sebestyen, N. A. Griffiths,

and G. Bisht. 2015. Representing northern peatland microtopography and hydrology

within the community land model. Biogeosciences 12:6463–6477.

Shumilova, O., D. Zak, T. Datry, D. von Schiller, R. Corti, A. Foulquier, B. Obrador, K. Tockner,

D. C. Allan, F. Altermatt, M. I. Arce, S. Arnon, D. Banas, A. Banegas-Medina, E. Beller,

1986       M. L. Blanchette, J. F. Blanco-Libreros, J. Blessing, I. G. Boëchat, K. Boersma, M. T.

Bogan, N. Bonada, N. R. Bond, K. Brintrup, A. Bruder, R. Burrows, T. Cancellario, S. M.

Carlson, S. Cauvy-Fraunié, N. Cid, M. Danger, B. de Freitas Terra, A. M. D. Girolamo,

R. del Campo, F. Dyer, A. Elosegi, E. Faye, C. Febria, R. Figueroa, B. Four, M. O.

Gessner, P. Gnohossou, R. G. Cerezo, L. Gomez-Gener, M. A. S. Graça, S. Guareschi,

B. Gücker, J. L. Hwan, S. Kubheka, S. D. Langhans, C. Leigh, C. J. Little, S. Lorenz, J.

Marshall, A. McIntosh, C. Mendoza-Lera, E. I. Meyer, M. Miliša, M. C. Mlambo, M.

Moleón, P. Negus, D. Niyogi, A. Papatheodoulou, I. Pardo, P. Paril, V. Pešić, P.

Rodriguez-Lozano, R. J. Rolls, M. M. Sanchez-Montoya, A. Savić, A. Steward, R.

Stubbington, A. Taleb, R. V. Vorste, N. Waltham, A. Zoppini, and C. Zarfl. 2019.

Simulating rewetting events in intermittent rivers and ephemeral streams: A global

analysis of leached nutrients and organic matter. Global Change Biology 25:1591–1611.

Siebert, S., F. T. Portmann, and P. Döll. 2010. Global patterns of cropland use intensity.

Remote Sensing 2:1625–1643.

Siev, S., E. C. Paringit, C. Yoshimura, and S. Hul. 2019. Modelling inundation patterns and

sediment dynamics in the extensive floodplain along the Tonle Sap River. River

Research and Applications 35:1387–1401.

Smith, A. P., B. Bond-Lamberty, B. W. Benscoter, M. M. Tfaily, C. R. Hinkle, C. Liu, and V. L.

Bailey. 2017. Shifts in pore connectivity from precipitation versus groundwater rewetting

increases soil carbon loss after drought. Nature Communications 8:1335.



Smith, J. A. M., K. J. Rossner, and D. P. Duran. 2021. New opportunities for conservation of a

rare tiger beetle on developed barrier island beaches. Journal of Insect Conservation

25:733–745.

Smyth, A. R., T. D. Loecke, T. E. Franz, and A. J. Burgin. 2019. Using high-frequency soil

oxygen sensors to predict greenhouse gas emissions from wetlands. Soil Biology and

Biochemistry 128:182–192.

Song, X., X. Chen, J. Stegen, G. Hammond, H. Song, H. Dai, E. Graham, and J. M. Zachara.

2018. Drought Conditions Maximize the Impact of High-Frequency Flow Variations on

Thermal Regimes and Biogeochemical Function in the Hyporheic Zone. Water

Resources Research 54:7361–7382.

Soupir, M. L., S. Mostaghimi, and C. E. Mitchem, Jr. 2009. A comparative study of stream-

gaging techniques for low-flow measurements in two Virginia tributaries. JAWRA Journal

of the American Water Resources Association 45:110–122.

Speir, S. L., J. L. Tank, and U. H. Mahl. 2020. Quantifying denitrification following floodplain

restoration via the two-stage ditch in an agricultural watershed. Ecological Engineering

155:105945.

Stallins, J. A., and A. J. Parker. 2003. The influence of complex systems interactions on barrier

island dune vegetation pattern and process. Annals of the Association of American

Geographers 93:13–29.

Stanford, J. A., M. S. Lorang, and F. R. Hauer. 2005. The shifting habitat mosaic of river

ecosystems. SIL Proceedings, 1922-2010 29:123–136.

Stanley, E. H., S. M. Powers, N. R. Lottig, I. Buffam, and J. T. Crawford. 2012. Contemporary

changes in dissolved organic carbon (DOC) in human-dominated rivers: is there a role

for DOC management? Freshwater Biology 57:26–42.

Stewart, B., J. B. Shanley, J. W. Kirchner, D. Norris, T. Adler, C. Bristol, A. A. Harpold, J. N.

Perdrial, D. M. Rizzo, G. Sterle, K. L. Underwood, H. Wen, and L. Li. 2022. Streams as



mirrors: Reading subsurface water chemistry from stream chemistry. Water Resources
Research 58.
Sullivan, S. M. P., M. C. Rains, and A. D. Rodewald. 2019. The proposed change to the
definition of "waters of the United States" flouts sound science. PNAS, 116:11558-

11561.

Sun, Z., L. Sandoval, R. Crystal-Ornelas, S. M. Mousavi, J. Wang, C. Lin, N. Cristea, D. Tong,
W. H. Carande, X. Ma, Y. Rao, J. A. Bednar, A. Tan, J. Wang, S. Purushotham, T. E.
Gill, J. Chastang, D. Howard, B. Holt, C. Gangodagamage, P. Zhao, P. Rivas, Z.
Chester, J. Orduz, and A. John. 2022. A review of Earth artificial intelligence. Computers
& Geosciences 159:105034.
Svensson, J. R., M. Lindegarth, P. R. Jonsson, and H. Pavia. 2012. Disturbance–diversity
models: what do they really predict and how are they tested? Proceedings of the Royal
Society B: Biological Sciences 279:2163–2170.
Sweet, W., J. Park, J. Marra, C. Zervas, and S. Gill. 2014. Sea level rise and nuisance flood
frequency changes around the United States.
Tagestad, J., N. D. Ward, D. Butman, and J. Stegen. 2021. Small streams dominate US tidal
reaches and will be disproportionately impacted by sea-level rise. Science of The Total
Environment 753:141944.
Thomas, M. A., B. B. Mirus, and J. B. Smith. 2020. Hillslopes in humid-tropical climates aren't
always wet: Implications for hydrologic response and landslide initiation in Puerto Rico.
Hydrological Processes 34:4307–4318.
Tiner, R. W. 2013. Tidal wetlands primer: An introduction to their ecology, natural history, status,
and conservation. University of Massachusetts Press, Amherst.
Tiner, R. W. 2017. Wetland indicators: A guide to wetland identification, delineation,
classification, and mapping. Second edition. Taylor & Francis, Boca Raton.



Tromp-van Meerveld, H. J., and J. J. McDonnell. 2006. Threshold relations in subsurface

stormflow: 2. The fill and spill hypothesis: Threshold flow relations. Water Resources

Research 42.

Tweedley, J. 2016. The contrasting ecology of temperate macrotidal and microtidal estuaries.
U.S. Geological Survey. 2017. Cottonwood Lake Study Area - Aerial Imagery: U.S. Geological

Survey data release, https://doi.org/10.5066/F7DZ06GR.

Valett, H. M., M. A. Baker, J. A. Morrice, C. S. Crawford, M. C. Molles Jr., C. N. Dahm, D. L.

Moyer, J. R. Thibault, and L. M. Ellis. 2005. Biogeochemical and metabolic responses to

the flood pulse in a semiarid floodplain. Ecology 86:220–234.

Van Meerveld, H. J. I., E. Sauquet, F. Gallart, C. Sefton, J. Seibert, and K. Bishop. 2020. Aqua

temporaria incognita. Hydrological Processes 34:5704–5711.

VanZomeren, C. M., J. F. Berkowitz, C. D. Piercy, and J. R. White. 2018. Restoring a degraded

marsh using thin layer sediment placement: Short term effects on soil physical and

biogeochemical properties. Ecological Engineering 120:61–67.

Venterink, H. O., N. M. Pieterse, J. D. M. Belgers, M. J. Wassen, and P. C. De Ruiter. 2002. N,

P, and K budgets along nutrient availability and productivity gradients in wetlands.

Ecological Applications 12:1010–1026.

Vorste, R. V., R. Corti, A. Sagouis, and T. Datry. 2016. Invertebrate communities in gravel-bed,

braided rivers are highly resilient to flow intermittence. Freshwater Science 35:164–177.

Waltham, N. J., and R. M. Connolly. 2011. Global extent and distribution of artificial, residential

waterways in estuaries. Estuarine, Coastal and Shelf Science 94:192–197.

Wang, X., W. Wang, and C. Tong. 2016. A review on impact of typhoons and hurricanes on

coastal wetland ecosystems. Acta Ecologica Sinica 36:23–29.

Wantzen, K., C. Alves, S. Badiane, R. Bala, M. Blettler, M. Callisto, Y. Cao, M. Kolb, G. Kondolf,

2081            M. Leite, D. Macedo, O. Mahdi, M. Neves, M. Peralta, V. Rotgé, G. Rueda-Delgado, A.



Scharager, A. Serra-Llobet, J.-L. Yengué, and A. Zingraff-Hamed. 2019. Urban stream

and wetland restoration in the Global South—A DPSIR analysis. Sustainability 11:4975.

Ward, J. V., K. Tockner, D. B. Arscott, and C. Claret. 2002. Riverine landscape diversity.

Freshwater Biology 47:517–539.

Ward, J. V., K. Tockner, and F. Schiemer. 1999. Biodiversity of floodplain river ecosystems:

ecotones and connectivity1. Regulated Rivers: Research & Management 15:125–139.

Ward, N. D., J. P. Megonigal, B. Bond-Lamberty, V. L. Bailey, D. Butman, E. A. Canuel, H.

Diefenderfer, N. K. Ganju, M. A. Goñi, E. B. Graham, C. S. Hopkinson, T. Khangaonkar,

2090        J. A. Langley, N. G. McDowell, A. N. Myers-Pigg, R. B. Neumann, C. L. Osburn, R. M.

Price, J. Rowland, A. Sengupta, M. Simard, P. E. Thornton, M. Tzortziou, R. Vargas, P.

B. Weisenhorn, and L. Windham-Myers. 2020. Representing the function and sensitivity

of coastal interfaces in Earth system models. Nature Communications 11:2458.

Watts, J. D., J. S. Kimball, A. Bartsch, and K. C. McDonald. 2014. Surface water inundation in

the boreal-Arctic: potential impacts on regional methane emissions. Environmental

Research Letters 9:075001.

Weird Bristol [@WeirdBristol] With a difference of 15-metres/49-foot between high and low tide,

the River Avon has the second largest tidal range in the world. Only the Bay of Fundy in

Canada has a higher tide, with an average of 16.8 metres/55-foot. #Bristol,

https://twitter.com/WeirdBristol/status/1015732213730758658; July 7, 2018.

Wen, H., J. Perdrial, B. W. Abbott, S. Bernal, R. Dupas, S. E. Godsey, A. Harpold, D. Rizzo, K.

Underwood, T. Adler, G. Sterle, and L. Li. 2020. Temperature controls production but

hydrology regulates export of dissolved organic carbon at the catchment scale.

Hydrology and Earth System Sciences 24:945–966.

Weyman, D. R. 1973. Measurements of the downslope flow of water in a soil. Journal of

Hydrology 20:267–288.



Whitworth, K. L., J. L. Kerr, L. M. Mosley, J. Conallin, L. Hardwick, and D. S. Baldwin. 2013.

Options for managing hypoxic blackwater in river systems: case studies and framework.

Environmental Management 52:837–850.

Wierzbicki, G., P. Ostrowski, and T. Falkowski. 2020. Applying floodplain geomorphology to

flood management (The Lower Vistula River upstream from Plock, Poland). Open

Geosciences 12:1003–1016.

Williams, D. D. 2006. The biology of temporary waters. Oxford University Press, Oxford ; New

York.

Wittenberg, H. 1999. Baseflow recession and recharge as nonlinear storage processes.

Hydrological Processes 13:715–726.

Wohl, E. 2021. An integrative conceptualization of floodplain storage. Reviews of Geophysics

59.

Wollheim, W. M., T. K. Harms, A. L. Robison, L. E. Koenig, A. M. Helton, C. Song, W. B.

Bowden, and J. C. Finlay. 2022. Superlinear scaling of riverine biogeochemical function

with watershed size. Nature Communications 13:1230.

Wu, B., F. Tian, M. Nabil, J. Bofana, Y. Lu, A. Elnashar, A. N. Beyene, M. Zhang, H. Zeng, and

2123        W. Zhu. 2023. Mapping global maximum irrigation extent at 30m resolution using the

irrigation performances under drought stress. Global Environmental Change 79:102652.

Wu, R., X. Chen, G. Hammond, G. Bisht, X. Song, M. Huang, G.-Y. Niu, and T. Ferre. 2021.

Coupling surface flow with high-performance subsurface reactive flow and transport

code PFLOTRAN. Environmental Modelling & Software 137:104959.

Xiao, D., Y. Shi, S. L. Brantley, B. Forsythe, R. DiBiase, K. Davis, and L. Li. 2019. Streamflow

Generation From Catchments of Contrasting Lithologies: The Role of Soil Properties,

Topography, and Catchment Size. Water Resources Research 55:9234–9257.

Xie, D., C. Schwarz, M. Z. M. Brückner, M. G. Kleinhans, D. H. Urrego, Z. Zhou, and B. Van

Maanen. 2020. Mangrove diversity loss under sea-level rise triggered by bio-



morphodynamic feedbacks and anthropogenic pressures. Environmental Research

Letters 15:114033.

Xin, P., A. Wilson, C. Shen, Z. Ge, K. B. Moffett, I. R. Santos, X. Chen, X. Xu, Y. Y. Y. Yau, W.

Moore, L. Li, and D. A. Barry. 2022. Surface water and groundwater interactions in salt

marshes and their impact on plant ecology and coastal biogeochemistry. Reviews of

Geophysics 60:e2021RG000740.

Zedler, P. H. 2003. Vernal pools and the concept of "isolated wetlands." Wetlands 23:597–607.
Zhang, Y. S., W. R. Cioffi, R. Cope, P. Daleo, E. Heywood, C. Hoyt, C. S. Smith, and B. R.

Silliman. 2018. A Global Synthesis Reveals Gaps in Coastal Habitat Restoration

Research. Sustainability 10:1040.

Zhang, Z., E. Fluet-Chouinard, K. Jensen, K. McDonald, G. Hugelius, T. Gumbricht, M. Carroll,

C. Prigent, A. Bartsch, and B. Poulter. 2021. Development of the global dataset of

Wetland Area and Dynamics for Methane Modeling (WAD2M). Earth System Science

Data 13:2001–2023.

Zhang, Z., N. E. Zimmermann, A. Stenke, X. Li, E. L. Hodson, G. Zhu, C. Huang, and B.

Poulter. 2017. Emerging role of wetland methane emissions in driving 21st century

climate change. Proceedings of the National Academy of Sciences 114:9647–9652.

Zhao, Y., X. Wang, S. Jiang, J. Xiao, J. Li, X. Zhou, H. Liu, Z. Hao, and K. Wang. 2022. Soil

development mediates precipitation control on plant productivity and diversity in alpine

grasslands. Geoderma 412:115721.

Zhi, W., and L. Li. 2020. The shallow and deep hypothesis: Subsurface vertical chemical

contrasts shape nitrate export patterns from different land uses. Environmental Science

& Technology 54:11915–11928.

Zimmer, M. A., A. J. Burgin, K. Kaiser, and J. Hosen. 2022. The unknown biogeochemical

impacts of drying rivers and streams. Nature Communications 13:7213.



Zimmer, M. A., K. E. Kaiser, J. R. Blaszczak, S. C. Zipper, J. C. Hammond, K. M. Fritz, K. H.

Costigan, J. Hosen, S. E. Godsey, G. H. Allen, S. Kampf, R. M. Burrows, C. A.

Krabbenhoft, W. Dodds, R. Hale, J. D. Olden, M. Shanafield, A. G. DelVecchia, A. S.

Ward, M. C. Mims, T. Datry, M. T. Bogan, K. S. Boersma, M. H. Busch, C. N. Jones, A.

2162        J. Burgin, and D. C. Allen. 2020. Zero or not? Causes and consequences of zero-flow

stream gage readings. WIREs Water 7.

Zimmer, M. A., and B. L. McGlynn. 2017. Ephemeral and intermittent runoff generation

processes in a low relief, highly weathered catchment. Water Resources Research

53:7055–7077.

Zipper, S. C., J. C. Hammond, M. Shanafield, M. Zimmer, T. Datry, C. N. Jones, K. E. Kaiser, S.

E. Godsey, R. M. Burrows, J. R. Blaszczak, M. H. Busch, A. N. Price, K. S. Boersma, A.

S. Ward, K. Costigan, G. H. Allen, C. A. Krabbenhoft, W. K. Dodds, M. C. Mims, J. D.

Olden, S. K. Kampf, A. J. Burgin, and D. C. Allen. 2021. Pervasive changes in stream

intermittency across the United States. Environmental Research Letters 16:084033.
