# Peer review of "Reviews and Syntheses: Variable Inundation # 2 Across Earth's Terrestrial Ecosystems"

_EGUsphere, 2024_

## Author Response (AR1)

Dear Editor,

Thank you for securing helpful reviews of our manuscript. Below we address each comment, with our response in bold text and reviewer suggestions in normal text. We have also submitted the revised manuscript. We greatly look forward to your further evaluation.

Sincerely,
James Stegen (on behalf of all co-authors)

########
Reviewer 1
General comments

This is an interesting review/opinion paper, which I understand resulted from a workshop. I feel it would stimulate thinking around the concept of VIE systems introduced by the authors. They provide useful pointers and mini-reviews on various types of VIEs as well as so ideas on how to improve monitoring and modelling of those, as well as ways to study them more holistically.

**Thank you for the encouraging remarks.**

As a paper capturing the proceedings of a workshop, its structure is more like a review/position paper than a research paper and I think it would be useful to present that structure more explicitly in the introduction to help the reader find their way. For example, some of the information in the section at lines 212-240 could be introduced earlier, thus giving more clarity.

**To address this we edited the last paragraph of the Introduction to let the reader know it is a 'Review and Synthesis' paper (note the title also states this) and to more explicitly summarize the structure of the manuscript. That paragraph now reads: "In this review and synthesis paper, we aim to catalyze cross-VIE science for the pursuit of transferable knowledge and ultimately models that are predictive across and aid in conserving contemporary and future VIEs. First, we briefly summarize high-level divergences in drivers of variable inundation, commonalities in the impacts of variable inundation, and then present expert mini-reviews of eight major VIE systems. These mini-reviews highlight that variable inundation occurs across vast ranges in spatial and temporal scales, which presents challenges to cross-VIE science. As such, we then overview these challenges and offer suggested solutions along with a summary of methods that are most relevant to VIE science. Finally, we conclude with perspectives on how cross-VIE science can derive transferable understanding to better protect these systems and their biodiversity."**

I made specific comments, which revolve for the most part around adding citations where it felt needed, and clarifying some points by adding examples or figures.

**Please see responses below.**

Scientific comments

P2 Introduction paragraph 1; the authors define a number of basic concepts for this paper. Some are general hydrology and could have a few key citations to back them up.

**References have been added to this paragraph, supporting all the concepts summarized.**

P2 Intro para 2; the authors define term VIE, and the conceptualisation around it. It would be useful to include a few key references that most likely informed the authors for this part.

**References have been added to this paragraph and the text has been revised to read: "Here, we define inundation as occurring when there is an uninterrupted aqueous barrier that limits diffusive gas exchange at the land-atmosphere interface (Elberling et al. 2011, Smith et al. 2018). This conceptualization includes diverse hydrological conditions ranging from free standing water to soil surface saturation. Hence, our broad definition spans from extreme events such as hurricane-driven flooding to shallow intermittent overland runoff across hillslopes. This definition does not attempt to separate 'inundation' from 'flooding' based on temporal frequency/duration, as has been proposed elsewhere (Flick et al. 2012)."**

P3 para 2; the authors give some examples of VIEs (including Fig1); it would be useful to state more explicitly that a VIE could belong to different types (especially, there is a wide range of human interventions, from light touch to heavy engineering so some systems may not be seen as human-engineered but would not be fully natural either).

**We edited the paragraph to now read: "Variable inundation occurs across a wide range of terrestrial ecosystems, but the factors governing its influences are typically studied independently without cross-ecosystem comparisons. Some examples of VIEs are hillslopes with overland flow, non-perennial streams, floodplains and parafluvial zones, variably inundated wetlands, vernal ponds/pools/playas, tidal systems, coastal systems impacted by storm-driven flooding, and human-engineered systems intended to shift inundation dynamics (e.g., flood-irrigated agriculture, stormwater infrastructure, and constructed wetlands) (Fig. 1). A given system may not fit clearly into a single VIE category and/or may transition across categories through time. For example, when flow ceases and isolated pools form in a non-perennial stream network, the stream begins to behave more like a wetland or vernal pond as opposed to a flowing stream. Further, while VIEs may be classified as wetlands under the broadest definition from the Ramsar Convention (Secretariat 2016), there is significant variation in how wetlands are defined (Finlayson and Van Der Valk 1995) and we do not attempt to rectify or clarify variation in those definitions. Here, when using the term 'wetland' we simply align with the perspective that wetlands are similar to marshes, swamps, and bogs."**

P3 line 98-109 interestingly, re point about intermittent rivers going permanent, there is a lot of research on how rivers are drying (so permanent rivers going intermittent), which looks like the

opposite problem. It could be useful to reflect about which issue could be the worst in terms of impacts.

**We edited the text to emphasize that inundation regimes are changing in multiple ways/directions. We prefer to avoid commentary on which type of change will have the most negative impacts as this is very much dependent on one's perspective and interests. A land manager responsible for fish populations may have a different perspective than a biogeochemist focused on carbon cycling, for example. The revised text reads: "Inundation dynamics are changing due to increased variability and magnitudes of precipitation and evapotranspiration, accelerated sea level rise, and human modifications to the Earth's land surface, including an increase in extreme events (Konapala et al. 2020, Li et al. 2022a). For example, extreme events such as coastal flooding are increasingly frequent (Vitousek et al. 2017). However, inundation patterns are changing in different ways across different VIEs (Zipper et al. 2021, Londe et al. 2022a). For example, in river systems seasonal drying is becoming more common in multiple biomes (Sweet et al. 2014, Zipper et al. 2021). While some rivers are shifting from non-perennial to perennial (Döll and Schmied 2012, Datry et al. 2018a) and others have fewer no-flow days than they did historically (Zipper et al. 2021). Divergence in the direction of change, with some systems inundating less and others inundating more, is likely linked to diverse drivers of change associated with changing climates and/or direct human impacts (Datry et al. 2023). Therefore, researchers and decision makers cannot rely exclusively on historical trends to predict future impacts (e.g., on species diversity) of changing inundation dynamics (Culley et al. 2016, Quinn et al. 2018, Rameshwaran et al. 2021, Li et al. 2022b)."**

P3 line 106-109; that sentence is not clear, and "dynamics" is repeated 3 times. Maybe you could split it in 2 sentences and clarify what the citations were actually about.

**The text has been edited and captured in the above paragraph.**

Fig 1 is good but for the middle world map pic marked (e); there doesn't seem to be an explanation for what VIE is (e).

**We removed the letter 'e' as it was only referencing the globe for the image credit statement. The image credit statement is updated accordingly, referring to the global image directly.**

P5 There is no citation in these 2 paragraphs. The first paragraph is a bit vague. First sentence mentions "models" but it is unclear of what. I guess it is models of impacts on VIEs. Some environmental variables are mentioned but without support from literature. Sentence on line 129-130 is a bit of a sweeping statement. The reference to VIE location in space is a bit confusing; at first, it is in terms of environmental variable space, then later it looks like the authors are talking about actual geographical location.

**The sentence referring to models has been edited to include a key reference and be more specific about the focal processes. It now reads: "Mechanistic knowledge that is transferable (per Schuwirth et al. 2019) across inundation regimes (i.e., from extreme events to predictable cycling) and across VIEs is required to develop hydrologic, biogeochemical, and ecological models that are predictive across contemporary and future conditions."**

**The sentences associated with VIE in 'space' have been edited to clarify we are referring to environmental space, not geographic location. In addition, these sentences are sweeping because they are a central tenet of the thesis of this manuscript. Those sentences now read: "Many other variables could be used, but regardless, environmental change will cause VIEs to shift within multi-dimensional environmental space. Predicting future impacts of variable inundation requires mechanistic understanding of how the location of a VIE in this multi-dimensional environmental space influences those potential impacts. We propose that our best chance to achieve such understanding is to generate knowledge of variable inundation impacts that is transferable across VIEs."**

P6 Fig2 the caption is very long and looks like some should be part of the main text.

**The caption has been shortened. We also confirmed that the concepts in the original version of the caption are contained in the main text. Trimming the caption does not, therefore, cause loss of conceptual content across the manuscript. The revised caption reads: "Figure 2. Conceptual overview of where different types of VIEs are often found within watersheds and some common shifts in system states across inundated and non-inundated conditions. VIEs are found from headwaters to coastal environments (Top) and the impacts of variable inundation have some consistencies across these diverse landscapes (Bottom). Organismal ecology, physiology, and demographics are altered by variable inundation, leading to shifts in community composition. Biogeochemical processes also shift, such as greater gas-phase transport of oxygen into soil/sediment when surface water is lost. A key goal for cross-VIE science is to mechanistically understand variation in the impacts of variable inundation across multi-dimensional environmental space. Credit: Nathan Johnson."**

P6-7 Lines 160-177 add key references to support this section.

**References have been added to this section.**

P7-8 I appreciate this section cannot cover every possible impact but I think it would be very useful to add a couple of sentences (maybe after line 211) about how flooding is actually part of the normal functioning of some ecosystems (eg river/floodplain connectivity for fish spawning, wetlands water level requirements for some bird species), in which case change in VIE regimes (timing, drying) can have huge impacts.

**The following sentence was added to the second paragraph of the introduction. This is within the paragraph that defines variable inundation. The higher position of this**

**sentence in the paper, we feel, increases the importance placed on the point that variable inundation is natural and even critical: "Variable inundation is natural in many systems and can be critical to system function (Shaeri Karimi et al. 2022, Tsoi et al. 2022), while in other systems it represents a disturbance (Sun et al. 2022a, Hopple et al. 2023)."**

Hillslope section. Re Fig. 3, the difference between 3b and 3c is not obvious looking at the pictures. Could the authors explain a bit more in the text where appropriate?

**The figure caption has been edited to be more explicit about the difference, which is no-flow ponding (b) and directional flow (c). In addition, the main text references each panel in context of a conceptual description. The revised figure caption reads: "Figure 3. Examples of variable inundation along hillslopes. a) looking downslope at an inundated slope; b) ponding with no flow due to microtopography; c) sheet wash with directional flow across the surface of a hillslope; d) rill formation with turbid water from erosion; e) vegetation community change on slope due to differences in soil moisture. All photos taken by Corianne Tatariw at Tanglewood Forest, Alabama."**

Hillslope section. P10 lines 304-309 Is there any citations that could support the statement on measuring hillslope VIE with earth observation? (IE examples of using EO for that purpose).

**References have been added. The text now reads: "Remote sensing could be used to identify and quantify these areas, spatially and temporally, based on sky-visible vegetation (e.g., plant morphologies, leaf nutrient contents) (Tai et al., 2020; Hwang et al., 2012) and topographic signatures (e.g. erosional patterns) (Trochim et al., 2015) caused by variable inundation."**

Non-perenial streams section. On the topic of monitoring intermittent streams, and routes to improve on that (eg UAVs), Dugdale et al. (2022) is useful to cite here (eg in P12 first paragraph). Dugdale et al. 2022. Looking to the skies: Realizing the combined potential of drones and thermal infrared imagery to advance hydrological process understanding in headwaters. Water Res Research 58

**We have included Dugdale et al., 2022 as a reference in the revised text, specifically in the final paragraph of this section, where we focus on challenges and opportunities for research in non-perennial streams. The text now reads: "Major challenges and opportunities include accurate mapping of non-perennial streams and accurate predictions of flow timing at annual, seasonal, and shorter time scales across scales. Headwaters, which are small, numerous, and often non-perennial (Kampf et al., 2021), are difficult to map and understand hydrologically, leading to knowledge gaps in the hydrological integrity of ecosystems at regional scales (Benstead and Leigh, 2012; Dugdale et al., 2022). While challenges remain, the use of drones and thermal infrared remote sensing could connect field observations with modeling to better understand the hydrology of these valuable systems (Dugdale et al., 2022)."**

Non-perennial streams section. It could be useful to elaborate briefly why these are undermonitored. For example, like headwaters, with which they partly overlap, intermittent rivers are undermonitored to a large extent because there are small and numerous, so the limited monitoring is more due to practicality and resources than anything else.

**The revised text now includes this information in the non-perennial stream section, as captured in our response to the previous comment.**

Non-perennial streams section. In section lines 386-398, it would be pertinent to cite Thompson et al. (2021) and their worldwide global warming/river flow alteration study. Thompson JR, Gosling SN, Zaherpour J, Laizé CLR. (2021) Increasing risk of ecological change to major rivers of the world with global warming. Earth's Future 9 (11), e2021EF002048

**The revised text now includes this citation and reads: "Thus, climate change is expected to lead to an increased risk of both high and low flows (Thompson et al, 2021)."**

Floodplain and fluvial section. Fig 5 On the biological drivers (left-hand side of main figure), beaver dams are listed; while it is relevant where there are beavers, it looks rather specific given the otherwise broad view taken for this paper.

**We changed the label to 'animal modifications' to be more general.**

Floodplain and fluvial section. Some paragraphs (eg 2nd) in this section doesn't have many references compared to the other mini-reviews, or compared to the other paragraphs. Please can you review if a few more key refs can be added. If it is that the text refers to the ones cited several times, then, can it be clear (it's fine citing them several times if needed).

**We added references for statements about controls on extent of flooding, rivers with extensive flooding (Amazon), and headwater rivers with little or no floodplain development.**

Floodplain and fluvial section. Another Thompson et al. (2021) ref would be useful to add (eg in 4th paragraph). Thompson JR, Laizé CLR, Acreman MC, Crawley A, Kingston DG. (2021). Impacts of climate change on environmental flows in West Africa's Upper Niger Basin and the Inner Niger Delta. Hydrology Research 52 (4), 958-974

**It is not apparent to us that this study of the effects of climate change is appropriate to add as a citation in this paragraph or elsewhere in the floodplain section.**

Human-engineered VIE section. I was wondering whether nature-based solutions would fall within this (eg, re-connecting river to upstream natural floodplain to mitigate flooding downstream).

**According to the US Department of the Interior, nature-based solutions "incorporate natural features and processes to protect, conserve, restore, sustainably use, and manage natural or modified ecosystems to address socio-environmental challenges**

**while providing measurable co-benefits to and benefit both people and nature." With this broad definition, nature-based solutions also include the management of existing natural systems. With this definition, we don't think it is appropriate to lump them in wholesale with human-engineered systems. However, we specify that systems engineered for restoration include those designed as nature-based solutions, and now mentioned them early in the vignette: "The primary drivers of human-engineered VIE formation explored here are land use change and restoration (including those for nature-based solutions), though hydrologic modifications impact inundation regimes of the natural VIEs explored earlier in the manuscript."**

**In addition, we included an acknowledgement at the end of the section that understanding baseline function for human-engineered VIEs can be beneficial for managing them as nature-based solutions: "A baseline understanding would also enable the restoration and repurposing of engineered VIEs as nature-based solutions (Clifford et al., 2023)."**

Inundation process and scale section (from P26 onward). Web links: the k26 weblink returns a dead link. Is there a primary source for the lake stats other than the Guinness? (ie a scientific source).

**We replaced the web links initially included in the paper with scientific papers: "Inundation volumes and surface areas of VIEs vary by at least sixteen orders of magnitude, from under $10^{-3}$ L to over $10^{13}$ L (Bonythan and Mason 1953), and $10^{-6}$ m$^2$ to over $10^{10}$ m$^2$ (Hess et al., 2015), respectively."**

Inundation process and scale section. This section is covering the important theme of scale, which can be quite complex to describe. I feel that it would benefit from a summary figure illustrating example of the different scales of organisms/processes, etc. to help the reader visualise better as they go through the text.

**We included a new figure highlighting the importance of scale in this section. The figure also helps define extent and grain in both time and space..**

Inundation process and scale section. Similarly to my point on intermittent river being undermonitored, it could be useful to elaborate briefly on why there is relative lack of monitoring or modelling at the finer scales, as there is probably some practical reasons, and if there are ways forward to improve (eg drones, Lidar).

**While we appreciate the suggestion to add information and details about the lack of monitoring in the scale section, we now address this issue in the Primary Methods section and have not further commented in the Process and Scale section to avoid repetition. The added text reads: "Fine-scale inundation dynamics, which have been historically hard to measure, can be captured using unmanned aerial vehicles (UAVs), which are often useful during or immediately after a significant inundation event such as flash flooding (Perks et al., 2016), to capture small-scale spatial dynamics that are**

difficult to detect with satellite or airborne methods (Dugdale et al., 2022, Manfreda et al., 2018), or to derive detailed data for input into hydrologic models and surface water calculations (Acharya et al., 2021)."

Primary methods section. Re monitoring, I was expecting mention of UAVs as they have potential to bridge between scales (eg between in-situ observation and traditional airborne and satellite EO. The Dugdale et al. (2022) paper I suggested above could actually be cited here.

**We made revisions to make it clear in the remote sensing portion of the Primary Methods section that multiple remote sensing techniques (satellite, drone, optical, microwave) are of use for monitoring VIEs. We have included Dugdale et al., 2022 as a reference in the Primary Methods section where we talk about the use of UAVs for inundation monitoring in addition to three more references that highlight the use of UAVs for hydrologic applications. The end of this paragraph now reads: "Fine-scale inundation dynamics can be captured using unmanned aerial vehicles (UAVs), which are often useful during or immediately after an inundation event such as flash flooding (Perks et al., 2016), to capture small-scale spatial dynamics that are difficult to detect with satellite or airborne methods (Dugdale et al., 2022, Manfreda et al., 2018), or to derive detailed data for input into hydrologic models and surface water calculations (Acharya et al., 2021)."**

Primary methods section. The second part of this section, on models, and particularly on ModEx is too vague. Could a concrete example of a ModEx approach be used to illustrate it and thus guide readers through the process, and/or a figure added.

**We focused revisions on the ModEx approach, which was within a single paragraph. We split that paragraph into two and added an example into the resulting second paragraph. It now reads: "In the context of VIEs, we expect ModEx to touch scales ranging from molecular microbiology to landscape ecology to regional ecosystem function to Earth system elemental cycles. As a landscape-scale example of ModEx, physical models could first be used to predict variable inundation across a watershed. Spatial and/or temporal uncertainty in those predictions could then be used to optimize collection of commercial remote sensing data. Those data would, in turn, be used to evaluate model predictions, leading to updated guidance from the model on where/when to collect additional remote sensing data. Further cycles could be pursued and model uncertainties could also guide collection of in situ data on variable inundation, organismal ecology, and/or biogeochemical processes. Many other examples across a variety of scales can be envisioned, and key to enabling this approach is the further development of models and measurement techniques that can capture system states in both inundated and non-inundated conditions. Techniques/models designed for specific kinds of ecosystems (e.g., perennial rivers) may be difficult to adapt. This emphasizes a need to do ModEx using models and measurements intentionally designed to span inundated and non-inundated system states."**

Cross-VIE transferrable knowledge section. Lots of interesting food for thought in this section. However, at times, it is a bit vague and would benefit from having concrete examples, or

specific citations. For example, in the first 2 paragraphs, the authors elaborate on the idea of using continuum approach, then they cite the wetlands unified classification; this could be detailed a bit further so that readers have an explicit example of what a continuum approach may look like.

**The first two paragraphs of this section have been revised to point towards tangible examples. The revised text now reads: "We propose that a key goal for VIE science is the development and open sharing of knowledge, models, algorithms, and data that transcend individual system types. Knowledge that crosses VIE systems will inherently span scales and levels of certainty from predictable, sub-daily inundation regimes to rare extreme events; integrating perspectives of these dynamic systems can aid in understanding and anticipating tipping points of physical, chemical, and biological components across VIEs. Development of such knowledge should be done via ModEx approaches coupled with ICON principles, which can generate models that can be used across VIEs. We suggest this can be facilitated through the development of conceptual models based on continuous environmental axes that modulate system responses to re-inundation (e.g., greenhouse gas production and changes in biological diversity).**

**Such continuum-based conceptual models necessitate going beyond discrete VIE categories by treating key physical characteristics as continuous variables that influence all VIE systems. One realization of such a conceptual model is summarized in Figure 12. Related approaches that are based on a suite of temporally variable ecological and geomorphological characteristics have proven useful for wetlands (Euliss et al. 2004, Lisenby et al. 2019). These wetlands frameworks have improved the understanding of human impacts on wetlands and led to more effective management (Wierzbicki et al. 2020, Mandishona and Knight 2022). These successes emphasize the potential effectiveness of continuum-based conceptual models for cross-VIE science."**

In paragraphs 3 and 4, slope and return period are presented as key control variables but there is no citation to really back this up.

**Several citations have been added to these paragraphs and throughout the broader section.**

With regards to the discussion about scale in the previous sections, how such an approach would capture different spatial scales (can slope do that? Or return period?)

**While slope doesn't directly relate to scale, the return internal axis is effectively the temporal scale of re-inundation, and this has now been emphasized in the revised text. When conducting studies across VIEs, spatial scale should be controlled or otherwise explicitly included in the study design; we added a sentence to highlight this point, which reads: "In doing so, we encourage careful attention towards the spatial and temporal scales of modeling and data generation efforts linked to return interval and slope."**

In addition, this section could be made clearer by shortening the text a bit (eg 1st paragraph could be more concisely worded).

**The first paragraph has been shortened and now reads: "We propose that a key goal for VIE science is the development and open sharing of knowledge, models, algorithms, and data that transcend individual system types. Knowledge that crosses VIE systems will inherently span scales and levels of certainty from predictable, sub-daily inundation regimes to rare extreme events; integrating perspectives of these dynamic systems can aid in understanding and anticipating tipping points of physical, chemical, and biological components across VIEs. Development of such knowledge should be done via ModEx approaches coupled with ICON principles, which can generate models that can be used across VIEs. We suggest this can be facilitated through the development of conceptual models based on continuous environmental axes that modulate system responses to re-inundation (e.g., greenhouse gas production and changes in biological diversity)."**

Technical comments

P2 L68 "variety of factors influence" should be "influences".

**We looked at this carefully and believe it should remain 'influence' because it is the 'influence' of multiple 'factors.'**

###########
Review 2
Initially I was skeptical of aggregating such complex and diverse systems into a model framework, but the eight VIE mini-reviews did an excellent job of showing similarities in ecosystem dynamics, microtopography, and modeling challenges.

**Thank you for the encouraging remarks.**

The mini-reviews seemed a little disjointed to me, it may help to organize these sections by discussing their components in a more connected way. For example, instead of stating fact after fact, discuss how "the knee bone is connected to the leg bone" throughout each of these ecosystems.

**When initiating the development of this manuscript we considered different ways of structuring the mini-reviews, recognized tradeoffs in different approaches, and finally settled on a common structure summarized in the manuscript. Generally, we view the mini-reviews as a way to briefly introduce different aspects of each type of ecosystem to provide context for later in the manuscript if readers are unfamiliar with a system outside the one they study. Further, we worked to provide information about each type of VIE that is somewhat distinguishing. We complemented this system-specific material with the first several paragraphs of the "Divergent Drivers, Common Responses, and VIE Mini-Reviews" section. That initial material provides an overview of some common elements and is more in the spirit of the reviewer's suggestion of writing about process**

**connections. Lastly, we feel that focusing on process connections within each VIE mini review could get repetitive as that cascade has some consistency across VIEs. We would, therefore, prefer to maintain the original structure.**

**Here is text from the manuscript summarizing the structure of each mini-review, which is aimed at helping the reader organize the information into consistent themes: "The following subsections present these mini-reviews which summarize system characteristics, drivers, and impacts of variable inundation with an emphasis on biogeochemistry and organismal ecology, and opportunities to better understand spatiotemporal patterns and impacts of variable inundation. Each mini-review is accompanied by a graphic that either provides a conceptual overview or imagery-based examples, with the goal of collectively touching on key drivers, dynamics, impacts, and tangible system examples."**

I can now see how a broad continuum utilizing ICON and ModEX methods may assist general VIE understanding.

**Thank you for the encouraging remarks.**

There are several places (ex. line 238-240) where comprehensive VIE models are proposed, but then never fully developed (excluding fig. 12 and the discussion on lines 876-890).

**We added a sentence following the referenced text to point the reader to the final section of the paper in which we propose the continuum perspective/approach. That added sentence reads: "This continuum perspective is developed as a conceptual model in the final section of the paper, titled "Towards Cross-VIE Transferable Understanding.""**

This work often felt like it was leading up to a big reveal of broad models such as the proposed topo/return interval model (fig.12), but then fell short at the end by stating a new encouraged perspective with some guidance. I think to address this head on it may potentially assist the paper to move the various model discussions (lines 992-1052) forward into the introduction.

**To address this we made the edit noted immediately above to point the reader to the final section. We also edited that final section to emphasize that the continuum perspective/approach is effectively a conceptual model. The conceptual model is summarized as Figure 12, which is based on quantitative axes and is used to propose hypotheses tied to greenhouse gasses and biological diversity. The text edits are throughout that section, so we do not include them directly in this response document. In addition, we prefer to keep the discussion of models in the section "Summary of Primary Methods used to Study VIEs" because the models are primarily discussed as tools and not conceptual models/frameworks.**

The two extent and granularity axes and examples (lines 876-891) discussed would benefit from a figure.

**A new figure has been added to the scale section. This new figure provides visual interpretation of extent and granularity for both space and time. The figure also includes panels that emphasize the nested nature of different scales observed by different 'agents' such as microbes, fish, and humans.**

There are also a few locations where the easily accessible term "climate change" is awkwardly substituted for distractingly complex language, the most notable instance occurs in the abstract. I suggest boldly using the term that is utilized throughout the rest of the writing.

**In the Abstract the associated sentence was edited to read: "We postulate that enhanced transferability will be important for predicting changes in VIE function in response to global change."**

Thank you for the opportunity to review!

**Thank you for your time and insights!**

############
Review 3
General comments

I greatly appreciate the careful outlining and meticulous preparation of this manuscript. I would have made different choices about the organization of the logic, but I review so few papers that are this carefully written these days that I just want to acknowledge how easy the majority of this paper is to read.

**Thank you for the encouraging remarks.**

While the logical progression of the paper is probably fine for some readers, my personal taste would suggest that the presented order of logic is missing a golden opportunity. Namely, I don't think the concepts of working with a "continuum perspective" in "multi-dimensional environmental space" is sufficiently developed before the mini review sections are presented. Furthermore, I don't think the mini review sections provide explicit enough references back to these themes to really let the paper fully illustrate the usefulness of the approach. I didn't really understand what "continuum perspective" meant until the very end of the paper, and I very much would have appreciated that understanding to provide a broader perspective with which to contextualize the mini reviews. As a result, I felt like this paper is in danger of supporting reductionist habits that are clearly intended to be avoided. Many of my specific comments are tied to this general criticism.

**There are many tradeoffs in how a manuscript is set up and some approaches will appeal more to some readers and less to others. While there is no perfect structure, we do appreciate the encouragement to think hard about ways to improve. After some**

**deliberation we'd like to propose adding a paragraph immediately before the first mini-review that briefly summarizes the conceptual model presented in Figure 12. The paragraph encourages the reader to start thinking in continuums instead of discrete types before reading the mini-reviews, providing context for the rest of the manuscript. We feel this helps address the reviewer's concern that readers will head into reductionist habits while also allowing the paper to build towards the conceptual model. The new paragraph reads: "This continuum perspective is more fully developed as a conceptual model in the final section of the paper, titled "Towards Cross-VIE Transferable Understanding." However, we briefly summarize here that it is based on two continuous environmental axes: inundation return interval and topographic slope. The concept is that these variables define a two-dimensional environmental space and that all VIE systems fall somewhere in that space. Impacts of variable inundation can be studied across this environment space instead of within discrete named types of VIEs. When going through the following mini-reviews, we encourage the reader to conceptualize each VIE type in context of return interval and slope (e.g., hillslopes may have a long return interval and steep slopes relative to tidal systems, while coastal systems inundated by storms may have similar slopes as tidal systems but much longer return intervals). The goal is to start viewing VIEs through a unified lens of environmental continuums."**

Specific comments

Line 70-72: The long history of general conceptual models from watershed hydrology that define infiltration excess (Hortonian) and saturation excess (Dunnian) overland flows may be relevant here. These concepts are quite abstract and do not imply specific mechanisms other than categorizing "top-down" or "bottom-up" sources of water. I realize that this is intended to be a more broadly applicable model, but a review of how existing concepts map to a transferable framework might be illustrative of its value.

**In response to this suggestion we made two edits. First, at the start of the Introduction we added additional citations, including to the classic Freeze 1974 review that summarizes Hortonian and Dunnian concepts. We also added a sentence pointing to these two flow concepts in the final section of the paper tied to the continuum-based conceptual model (Figure 12). This new sentence reads: "We may learn that additional axes are needed and these may be linked to other conceptual models, such as whether inundation emerges through infiltration-excess (Hortonian flow generation) or through saturation-excess (Dunnian flow generation) (Freeze 1974)." We feel this adds a helpful element of pointing out that there are other conceptual models and that they may prove important to integrate into our framework. By keeping the edits brief, we also avoid a significant increase in length and complexity of the manuscript.**

Line 75: While I suppose it is implied by preceding statements, this paragraph seems to imply that the control volume for the water budget terms used is strictly associated with surface water? That must be true for "export via infiltration" in this sentence to make sense. I wonder if

the control volume should be more explicitly defined to avoid the reader having to guess at the interpretation of the water budget in this paragraph.

**We acknowledge that groundwater dynamics are important and influence surface water. In this manuscript we focus on surface water as the realization of inundation, per our definition. We edited text to help clarify this in the opening lines of the paragraph, which now read: "A variety of factors influence surface water retention, infiltration, flow, and surface expression within an ecosystem, such as land surface relief, topographic slope, subsurface permeability, evapotranspiration, and human-based modifications of the landscape." In addition, we clarify this point again at the end of the paragraph by stating, "Regardless of where water comes from, its expression at the land-atmosphere interface occurs when the rate of water supply is greater than the rate of export via infiltration, evapotranspiration, and runoff."**

Line 76: Would it be accurate to say "… continuous aqueous barrier of surface water…" to clarify the control volume per the previous comment?

**The first three sentences of this paragraph have been edited in the spirit of this suggestion and now read: "Here, we define inundation as occurring when there is an uninterrupted aqueous barrier that limits diffusive gas exchange at the land-atmosphere interface (Elberling et al. 2011, Smith et al. 2018). This conceptualization includes diverse hydrological conditions ranging from free standing water to soil surface saturation. Hence, our broad definition spans from extreme events such as hurricane-driven inundation to shallow intermittent overland runoff across hillslopes."**

Line 130: I can only guess at what "multi-dimensional environmental space" means at this point. I didn't really understand what it meant in the abstract, either. Does it reference simply the dimensions of space and time? If not, what other dimensions are implied? This seems like buzz-wordy jargon without a little more development.

**We use multi-dimensional environmental space here to represent the multiple variables that influence an ecosystem; including its biogeochemical reactions and organisms (similar to multidimensional niche space). We edited the text and included more examples to help with clarity. The text now reads: "We envision the impacts of variable inundation as dependent on the location of any given VIE within multi-dimensional environmental space. This space can be defined with a variety of environmental variables such as inundation return interval and duration, topographic slope, geology, vegetation composition, precipitation, salinity, and temperature. Similar to multi-dimensional niche space (Hutchinson 1978), many other variables could be used, but regardless, environmental change will alter the position of VIEs within continuous, multi-dimensional environmental space."**

Line 238: The term "continuum perspective" has been used a couple times but has yet to be explicitly defined. Is this in reference to the continuum of the flow of water mentioned in introducing the organization of mini-reviews, or some other sort of continuum? Without an

explicit definition, I have a hard time agreeing that a "continuum perspective" will somehow bridge the conceptual models of variable inundation across many different types of these systems.

**We added a new paragraph just before the first system vignette to help define the continuum perspective. The first part of that paragraph reads: "This continuum perspective is more fully developed as a conceptual model in the final section of the paper, titled "Towards Cross-VIE Transferable Understanding." However, we briefly summarize here that it is based on two continuous environmental axes: inundation return interval and topographic slope.  These variables can be used to define a two-dimensional environmental space  that contains all VIE systems."**

Lines 237 to 240: After reading the first 2 mini reviews, I have come back to make this comment. As of now, I have not seen any specific references to the "continuum perspective" or any specific references to the changes of the systems in "multi-dimensional environmental space". So far, these terms have still not been defined, nor have specific examples of their application helped clarify how they are useful in promoting transferable concepts. I would highly recommend that the mini reviews use these terms directly and provide some specific examples of how the behavior of that specific system lends to the transferable perspective being promoted. Lines 232-235 seems to be asking the reader to figure out these connections for themselves, without any guiding vision for this exercise provided by the paper. I am deep in the paper and am still not really sure how it is promoting transferable conceptual models, other than I am supposed to somehow be thinking about a "continuum" or about a change in "multi-dimensional space".

**Please see our response to the comment immediately above. The added paragraph is meant to address this reviewer suggestion as well.**

Line 504: More information here for those not familiar with the location of the Pantanal? If the goal is to encourage cross-domain thinking, less assumption of domain-specific knowledge is critical.

**We agree that having less domain-specific knowledge is essential for a manuscript in which we are advocating for cross-discipline thinking. The text has been edited to read**: **"While the largest variably inundated wetlands are connected to floodplains, like the 130,000 km$^2$ Pantanal located in Brazil and extending into Bolivia and Paraguay (Ivory et al. 2019), non-floodplain wetlands surrounded by upland (also known as geographically isolated wetlands) as large as ~6 ha may also experience whole-system drying and rewetting (Lane and D'Amico 2016)."**

Line 611 to 613: The sentence "As inundation regimes may become more variable, increasing conservation and protection efforts for ephemeral and temporary ponds may become more essential to maintain these critical VIEs" seems circular. Perhaps "As inundation regimes may become more variable, increasing conservation and protection efforts for maintaining ephemeral and temporary pond VIEs may become more essential."

**We edited the sentence to improve readability, and it now reads: "As inundation regimes become more variable, increasing conservation and protection efforts for maintaining ephemeral and temporary ponds will become more essential."**

Line 632: The "area of coastal inundation" rather than the "size of coastal inundation"? Not clear what dimension "size" means in this context.

**We made the suggested change, and the text now reads: "The impact and areal extent of coastal inundation varies across events, depending on topography, infrastructure, and event size (Fig. 8)."**

Line 755-756: Is "three-dimensional physical space" part of "multi-dimensional environmental space"? This is the first mention of something approaching this bridging concept in the mini reviews.

**This sentence has been edited and now reads: "Similarly, complex feedbacks exist among hydrology, biogeochemistry, ecology, and geomorphology (Xin et al. 2022); these dynamics may need to be considered in future ecosystem projections."**

Line 766 and Figure 9: I find this figure hard to follow other than the gradients at the bottom. Are the dashed arrows about groundwater or the distance of inundation? Showing them below the land surface may be misleading. The crab seems like a visual non-sequitur.

**We edited the figure caption to increase clarity of the meaning of dashed arrows and the crab depiction, which now reads: "*Tidally driven coastal zones span sediments exposed at low tide to marshes and coastal forests inundated at high tide. This lateral gradient of tidal exposure across micro to macro-tidal systems (dotted black lines), alters physical (e.g., particle deposition), biological (e.g., species composition), and chemical (e.g., nutrient transformations) factors. Organisms can impact conditions along the gradient, such as flow path alteration by crab burrowing. Credit: Nathan Johnson.*"**

Line 781-785: This list is difficult to read and appears to mix punctuation with semicolons and commas.

**We edited the text to be more consistent, which now reads: "Examples of land-use driven human-engineered VIEs include, but are not limited to: croplands irrigated to the point of inundation (e.g., rice paddies, cranberry bogs), canals for irrigation, drainage and stormwater (e.g., roadside ditches, retention ponds), and unintentional VIE formation following landscape modification (e.g., "accidental" urban wetlands (Palta et al. 2017) and ponds in agricultural fields (Saadat et al. 2020)."**

Line 865: This paragraph is suddenly using lakes and rivers of endorheic basins as an example of large scale variably inundated areas when none of the mini reviews provides a description of this particular example. This lack of connection seems a bit awkward.…

**We agree with the reviewer that these examples require more of a transition than what exists in the main text. We edited the opening of this paragraph to provide a clearer connection, which now reads: "VIEs span broad spatiotemporal scales of variable inundation, from small wetlands and vernal ponds to the floodplains of the world's largest rivers. While the examples in the mini-reviews focus on eight different ecosystems, variably inundated ecosystems are even broader such as mosses and pore spaces that are periodically covered by droplets of water and vast endorheic lakes and rivers."**

Line 1063: This is the first time the idea of a "continuum approach" has been revisited since being introduced (without being defined) before the mini reviews. Also, figure 12 is the first concrete example of what is meant by "continuum approach", and none of the ideas in figure 12 are actually discussed in the narrative of the paper. Should those ideas for some sort of integrating model be relegated exclusively to a figure caption? If the reductionist habits of categorizations used for the mini reviews is something to be avoided, why isn't that being reinforced with visions for how to avoid it through the whole paper? Why is the conceptual model of figure12 not presented sooner, such that we can think about where each type of system fits as we read each mini review? That would make the mini reviews much more interesting to read and less of a list of loosely categorized details.

**This comment is highly related to the next comment and we've combined our response under the next comment.**

Line 1080-1089: I would have loved to have been explicitly informed that these were hypothesized to be the foundation of at least one "continuum" concept before reading the mini review sections. Then I would have liked if the mini review sections to explicitly discuss how our current understanding of each type of VIE feeds these overarching ideas.

**Our primary approach to address this comment and the previous comment is via the addition of a new paragraph immediately prior to the first vignette, and editing the preceding paragraph. Together these paragraphs provide a roadmap so the reader understands the arc of the paper and gets their mind thinking towards the continuum perspective. Those two paragraph's read: "We separate VIEs into categories as a heuristic simplification that allows for an appreciation of variation and commonalities in drivers, impacts, and opportunities. We anticipate that the disciplinary foci of individual researchers will align most closely with a subset of the summarized VIE types. One goal of this manuscript is to facilitate researchers thinking about how their science applies across multiple VIEs. We emphasize that in many (and maybe all) cases there is not a clear distinction among the types of VIEs we discuss below (e.g., non-perennial streams can be inundated due to storm surge, resulting in floodplains or parafluvial zones). Ultimately, we encourage a continuum perspective that does not rely on discrete system names or hard boundaries, and instead views VIEs across multi-dimensional**

**environmental space based on inundation regimes and physical settings such as topographic slope.**

**This continuum perspective is more fully developed as a conceptual model in the final section of the paper, titled "Towards Cross-VIE Transferable Understanding." However, we briefly summarize here that it is based on two continuous environmental axes: inundation return interval and topographic slope.  These variables can be used to define a two-dimensional environmental space  that contains all VIE systems. With this model, impacts of variable inundation can be studied across environment space instead of within discrete named types of VIEs. When going through the following mini-reviews, we encourage the reader to conceptualize each VIE type in context of return interval and slope (e.g., hillslopes may have a long return interval and steep slopes relative to tidal systems, while coastal systems inundated by storms may have similar slopes as tidal systems but much longer return intervals). When VIEs are viewed through a unified lens of environmental continuums, larger interdisciplinary questions may be answered."**

Technical corrections

Line 953: The phrase  "...revealing the governing the processes…" appears to be missing some words?

**The sentence was edited to read: "Conversely, larger scale measurements integrate across finer-scale processes to quantify ecosystem dynamics and properties, but without necessarily revealing what governs those processes."**

Line 966: Suggest "…captured by a passive microwave radiometer as well as through C- and L-band radar backscatter…"

**The suggested edits were made.**